# The splicing regulator PTBP2 controls a program of embryonic splicing required for neuronal maturation

Qin Li[1], Sika Zheng[1], Areum Han[2], Chia-Ho Lin[1], Peter Stoilov[3], Xiang-Dong Fu[4], Douglas L Black[1,5]*

[1]Department of Microbiology, Immunology and Molecular Genetics, University of California, Los Angeles, Los Angeles, United States; [2]Department of Bioengineering, Henry Samueli School of Engineering and Applied Science, University of California, Los Angeles, Los Angeles, United States; [3]Department of Biochemistry, West Virginia University, School of Medicine, Morgantown, United States; [4]Department of Cellular and Molecular Medicine, University of California, San Diego, La Jolla, United States; [5]Howard Hughes Medical Institute, University of California, Los Angeles, Los Angeles, United States

**Abstract** We show that the splicing regulator PTBP2 controls a genetic program essential for neuronal maturation. Depletion of PTBP2 in developing mouse cortex leads to degeneration of these tissues over the first three postnatal weeks, a time when the normal cortex expands and develops mature circuits. Cultured *Ptbp2*[−/−] neurons exhibit the same initial viability as wild type, with proper neurite outgrowth and marker expression. However, these mutant cells subsequently fail to mature and die after a week in culture. Transcriptome-wide analyses identify many exons that share a pattern of mis-regulation in the mutant brains, where isoforms normally found in adults are precociously expressed in the developing embryo. These transcripts encode proteins affecting neurite growth, pre- and post-synaptic assembly, and synaptic transmission. Our results define a new genetic regulatory program, where PTBP2 acts to temporarily repress expression of adult protein isoforms until the final maturation of the neuron.

*For correspondence: dougb@microbio.ucla.edu

**Reviewing editor**: Benjamin J Blencowe, University of Toronto, Canada

## Introduction

Compared to other cells, neurons undergo an unusually long period of maturation as they differentiate. In rodents, the time required for neuronal progenitors to become fully functional neurons can be many days, during which cells will exit mitosis, migrate to a proper location, grow initial neurites to be defined in their polarity, fully extend these axons and dendrites to their proper targets, and finally assemble active synaptic connections and circuits (*Waites et al., 2005*; *Craig et al., 2006*; *Hamby et al., 2008*; *Barnes and Polleux, 2009*). A wide variety of genetic mechanisms ensure that these events occur in proper sequence and induce cell death when a process is incorrect. This developmental pathway is best understood at the level of transcriptional regulation where important signaling molecules, transcription factors, and epigenetic modifications are implicated in lineage commitment, cell survival, and establishment of synaptic connections. However, many aspects of the genetic control of neuronal development are not understood.

Alternative pre-messenger RNA splicing is a common mechanism for genetic control in the nervous system, where many proteins important for neuronal differentiation and function are made in diverse isoforms through changes in splice site choice (*Licatalosi and Darnell, 2006*; *Li et al., 2007*; *Norris and Calarco, 2012*; *Yap and Makeyev, 2013*; *Zheng and Black, 2013*). Changes of splicing pattern

**eLife digest** Cells within the developing brain undergo an extended period of maturation. A neuronal progenitor cell must first migrate to the proper place within the brain and then develop long extensions that become the axon and dendrites used by the mature neuron to communicate with other cells. Finally, the synapses that connect neurons with other neurons must be established. Multiple mechanisms are needed to ensure that all the proteins involved in this process are expressed when and where they are needed.

The production of a protein begins with a region of DNA being transcribed to produce an RNA transcript that consists of segments called exons separated by segments called introns. This transcript then undergoes a process called splicing that involves the introns being removed and the exons being joined together to form a messenger RNA molecule that can be translated into protein. Specialized RNA binding proteins regulate the splicing process, and most RNA transcripts are subject to a form of splicing called alternative splicing that allows a single gene to express more than one messenger RNA molecule and hence more than one protein product.

As neuronal progenitor cells in the brain are induced to mature into neurons, many RNA transcripts are seen to change their splicing patterns. At the same time, the level of a regulatory RNA binding protein called PTBP1 decreases and the level of a related protein called PTBP2 increases. Now Li et al. have studied mutant mice that lack PTBP2, and have found that structures of the forebrain that normally undergo extensive development after birth instead experience tissue degeneration when PTBP2 is absent. Similarly, when neurons lacking PTPB2 are grown in culture, they fail to develop correctly and die.

Li et al. also found that messenger RNAs from many genes involved in postnatal brain development—affecting processes such as the growth of axons and dendrites and the formation of synapses—exhibit defective alternative splicing in the mutant mice. Specifically, protein variants that would normally be expressed only in adult brains were being expressed much earlier.

By inhibiting the expression of adult forms of proteins until neurons have matured, PTBP2 plays an essential role in controlling the brain's early development. Further work is now required to determine how individual changes in messenger RNA and protein structure controlled by PTBP2 might alter protein function between immature and mature neurons.

are regulated by *trans*-acting RNA-binding proteins that can be expressed in a temporal- or cell-type-specific manner to enhance or silence particular splicing events (*Black, 2003*; *Matlin et al., 2005*; *Chen and Manley, 2009*; *Kalsotra and Cooper, 2011*; *Braunschweig et al., 2013*; *Kornblihtt et al., 2013*). Splicing regulators generally control large numbers of target transcripts and several have been shown to be essential for proper brain development or function (*Jensen et al., 2000*; *Gehman et al., 2011, 2012*; *Yano et al., 2010*; *Iijima et al., 2011*; *Charizanis et al., 2012*; *Ince-Dunn et al., 2012*).

At the onset of neuronal differentiation, there is a change in the expression of two polypyrimidine tract binding proteins, PTBP1 (also known as PTB or HnRNP I) and PTBP2 (also known as nPTB or brPTB) (*Keppetipola et al., 2012*). Early in development, PTBP2 is present at low levels in neuronal progenitor cells (NPC), but is sharply upregulated when NPCs are induced to differentiate into post-mitotic neurons (*Boutz et al., 2007*; *Tang et al., 2011*; *Zheng et al., 2012*). In contrast, PTBP1 is expressed at high levels in neural progenitor cells, glia, and other non-neuronal cells, but is repressed in differentiating neurons coincident with the increase in PTBP2 concentration (*Polydorides et al., 2000*; *Boutz et al., 2007*; *Makeyev et al., 2007*; *Zheng et al., 2012*). The repression of PTBP1 is mediated by the miRNA miR124 (*Makeyev et al., 2007*), which leads to the induction of PTBP2 expression through additional post-transcriptional mechanisms (*Boutz et al., 2007*; *Makeyev et al., 2007*; *Spellman et al., 2007*). SiRNA mediated depletion of PTBP1 stimulates PTBP2 expression and can be sufficient to induce neuronal differentiation of fibroblasts (*Xue et al., 2013*). However, the role of PTBP2 in differentiating neurons is not understood.

PTBP1 and PTBP2 are encoded by paralogous genes and are very similar in peptide sequence, especially in their four RNA recognition motifs (RRMs), giving them similar RNA-binding properties (*Oberstrass et al., 2005*). The best-characterized function of the two PTB proteins is in the regulation of alternative splicing patterns. PTBP1 and PTBP2 bind to CU-rich regulatory sequences within or

adjacent to many alternative exons to repress or activate their splicing, or sometimes to cause intron retention (*Xue et al., 2009*; *Llorian et al., 2010*; *Kafasla et al., 2012*; *Keppetipola et al., 2012*). However the two proteins differ in their splicing regulatory activities. Some exons, such as exon 8A of the CaV1.2 calcium channel transcript (*Cacna1c*), are more strongly repressed by PTBP1 than PTBP2, whereas other exons, such as exon 18 of the PSD95 transcript (*Dlg4*), are affected by both proteins (*Markovtsov et al., 2000*; *Tang et al., 2011*; *Zheng et al., 2012*). Neither the mechanisms underlying the different responses of exons to the two PTB proteins, nor the roles of their partially overlapping splicing programs in neuronal development are known.

To identify roles of PTBP2 in the developing mouse brain, we used conditional gene targeting. We show that the loss of PTBP2 leads to a catastrophic failure in neuronal maturation and to the misexpression of many protein isoforms affecting neurite growth, synapse formation and synaptic transmission. We find that PTBP2 and the program of splicing it controls are critical to both embryonic and postnatal brain development.

## Results

### Loss of PTBP2 in all neurons has limited effects on brain morphology but leads to lethality at birth

PTBP2 is induced as nestin-positive neuronal progenitors exit mitosis and begin to differentiate (*Boutz et al., 2007*). This can be seen in the embryonic cortex and olfactory bulb where proliferating cells in the subventricular zone are stained for nestin, but more mature cells populating the outer cortical layers have lost nestin expression but gained PTBP2 immunoreactivity (*Figure 1A*). To investigate the function of PTBP2 in the developing brain, we generated a conditional null allele carrying loxP sites flanking exon 4 of the *Ptbp2* gene (*Ptbp2^loxP^*) (*Figure 1B*). In targeted tissues, Cre-mediated deletion of exon 4 introduces a reading frame shift to render the allele null. Correct targeting in ES cells and germ line transmission were confirmed by Southern blot and PCR of genomic DNA (*Figure 1C,D*). A *Ptbp2* null allele (*Ptbp2^−^*) was generated in crosses of *Ptbp2^loxP/+^* mice to a germline-active Cre transgenic line *EIIaCre* (*Lakso et al., 1996*). Homozygous *Ptbp2* null pups generated in subsequent crosses failed to initiate breathing and died due to apparent respiratory failure at birth (*Figure 1E*). Unlike their wild-type or heterozygous littermates, they do not respond to touch and appear completely paralyzed. Immunoblots of E18.5 whole brain lysates confirm the complete absence of PTBP2 in homozygous mutant mice and reduced protein in heterozygous mice compared to wild type (*Figure 1F*). We also do not observe a truncated PTBP2 protein arising from the mutant allele (*Figure 1—figure supplement 1*). This phenotype is consistent with that of another full *Ptbp2* null allele (*Licatalosi et al., 2012*). Since *Ptbp2* is also expressed in cardiac and skeletal muscle, defects in these tissues may contribute to the paralysis and death of the germline mutant mice.

To examine PTBP2 function specifically in the nervous system, *Ptbp2^LoxP/LoxP^* mice were crossed to a *Nestin*-Cre transgenic line, expressing Cre in all neuronal progenitor cells, and bred to obtain homozygous *Ptbp2^LoxP/LoxP^/Nestin-Cre^+/−^* (*Ptbp2*-NesKO) mice (*Tronche et al., 1999*). Unlike the full null, these pups initiated breathing, although at a lower rate than wild type and appeared cyanotic, dying within 1 hr after birth. Immunoblots of E18.5 whole brain from *Ptbp2*-NesKO pups confirmed the absence of PTBP2 protein (*Figure 1F*, left two lanes). Both *Ptbp2* null and *Ptbp2*-NesKO embryos were recovered at Mendelian ratios (27 out of 91, and 12 out of 50) at the late embryonic stage E18.5, suggesting that the lethality was specific to neonates.

Immunohistochemistry with PTBP2 antibody on sections of E18.5 *Ptbp2^LoxP/LoxP^/Nestin-Cre* brain showed that the vast majority of the neurons in the CNS had lost PTBP2 expression (*Figure 2A*), while PTBP2 expression in heart and skeletal muscle was unaffected (data not shown). Expression of *Ptbp1*, a close paralog of *Ptbp2* expressed in neural progenitor cells as well as astrocytes, ependymal cells, and other non-neuronal cell types in the brain, was not changed in the absence of PTBP2 (data not shown). This suggests that, although PTBP1 can repress PTBP2 expression in cells where PTBP1 is predominant, PTBP2 is not required to maintain PTBP1 repression in post-mitotic neurons where PTBP2 predominates. These results demonstrate that neonatal survival requires PTBP2 expression specifically in the CNS.

To assess if loss of PTBP2 leads to developmental defects in the brain, we examined *Ptbp2*-null brains at late embryonic stage E18.5. When stained by Nissl or hematoxylin and eosin (H&E), mutant embryos showed largely normal morphology for major brain structures such as the neocortex, striatum, hippocampus, and thalamus (*Figure 2BC*). The neocortex showed normal thickness and laminar

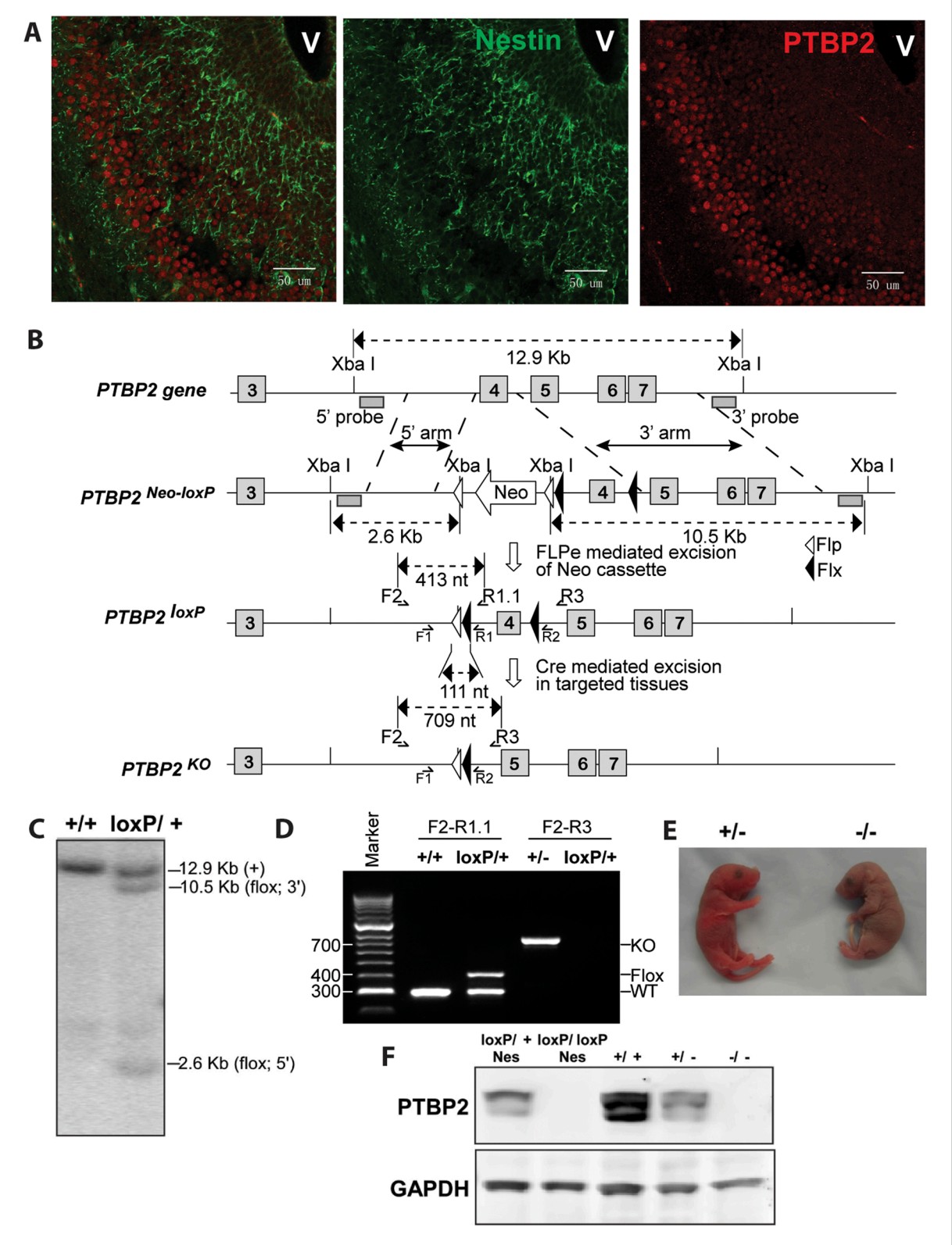

**Figure 1**. Generation and validation of the *Ptbp2* conditional mutation. (**A**) Expression of PTBP2 in maturing neurons of E15.5 brain. Nestin-positive cells (green, middle panel) of the subventricular zone express only limited PTBP2 (red, right panel), but as these cells mature and migrate to outer layers, PTBP2 is induced (Overlay panel to the left). V indicates the ventricle. (**B**) The targeting construct carrying loxP sites flanking exon 4 of *Ptbp2* was

*Figure 1. Continued on next page*

Figure 1. Continued

integrated into the endogenous *Ptbp2* locus by homologous recombination. The neomycin (Neo) selection cassette flanked by Frt sites was removed by crossing founder mice (*Ptbp2*[Neo-loxP/+]) with 129S4/SvJaeSor-Gt(ROSA)26Sortm1(FLP1)Dym/J mice (Jackson) carrying Flp recombinase. The resulting *Ptbp2*[loxP/+] mice were bred to C57Bl/6 Cre transgenic strains to generate *Ptbp2*[+/−] animals in targeted tissues (**C** and **D**). Genotypes were verified by Southern blot of genomic DNA digested with XbaI (**C**), and by PCR (**D**). (**E**) *Ptbp2* null mice display cyanosis and die immediately after birth due to respiratory failure. (**F**) Immunoblot for PTBP2 confirms that the disrupted allele eliminates expression of PTBP2.

The following figure supplements are available for figure 1:

**Figure supplement 1**. Immunoblot of PTBP2 in Emx-knockout and wild-type brain. Truncated proteins are not observed.

organization (*Figure 2B*). Immunostaining for a variety of neuronal markers, including pan-neuronal markers (Map2, TuJ1), interneuron subtype markers (calbindin and calrectinin), a marker for proliferating nuclei (phosphoHistone3), and for glial and radial glial markers (GFAP and RC2) all showed no significant differences between mutant and control littermate brains (data not shown). To assess possible defects in neuronal migration, we examined cortical layer markers (Tbr2, Brn2 and Cux1) and again did not observe any significant differences between mutant and control brains (data not shown). Notably, we did not observe ectopic subventricular mitotic cells in either the full null or the *Ptbp2*-NesKO brains. Such cells were previously reported as sporadically occurring in another full germline *Ptbp2* null mouse (*Licatalosi et al., 2012*). These different observations may result from different sensitivity in staining methods or differences in genetic background.

Although the grey matter of the *Ptbp2* mutant brain appears largely normal in its overall patterning and the specification of major neuronal subtypes, examination of Nissl stained coronal sections revealed some white matter defects (*Figure 2C*). Several major axonal tracts were either absent or significantly diminished in the *Ptbp2* mutant brain; the internal capsule, external capsule, and the lateral olfactory tract were reduced, and the anterior commissure was absent in the *Ptbp2* mutant. These major axonal tracts connect distant regions of the developing brain, such as cortex, thalamus, and the brainstem, and are essential to basic physiological functions such as respiratory control. These axonal defects may thus contribute to the neonatal lethal phenotype. However, quantification of this phenotype and linking it to neonatal death will require more extensive analysis of mice at a variety of developmental timepoints. These observations are consistent with a role for PTBP2 in axonogenesis or myelination (See *Figures 3 and 4* below).

## Inactivation of *Ptbp2* in higher forebrain causes postnatal lethality and early neuronal death in culture

PTBP2 expression goes down during the first postnatal week of brain development with moderate levels maintained into adulthood. Identified PTBP2 targets such as PSD95 (*Dlg4*) show changes in splicing coincident with this postnatal depletion of PTBP2 (*Zheng et al., 2012*). Because of their neonatal lethal phenotype, the postnatal effects of PTBP2 loss cannot be examined in either the germline or pan-CNS null mice. To look at PTBP2 function later in development, we crossed *Ptbp2*[loxP/loxP] mice with *Emx-Cre* transgenic mice, where Cre recombinase is expressed in projecting neurons of the higher forebrain, allowing selective gene disruption in the cerebral cortex, hippocampus, and olfactory bulb (*Iwasato et al., 2000*). In *Ptbp2*[loxP/loxP]:*Emx1-Cre*[+/−] mice (*Ptbp2*-EmxKO), *Ptbp2* is highly depleted from the cerebral cortex, hippocampus, and olfactory bulb, but still expressed at high levels in other brain regions, such as cerebellum, striatum, and brain stem (*Figure 3AB* and data not shown).

Unlike the Nestin and Germline-Cre knockouts, *Ptbp2*-EmxKO pups were viable, with body weights similar to control newborns at P0. However, these mutant mice displayed slower growth than their littermates and could be identified by their small size and low body weight as early as postnatal day 3 (*Figure 3A*). By P14, their average weight was less than half of control pups, and they all died around weaning age (P18-21). The cause of the *Ptbp2*-EmxKO pups' failure to thrive was apparent upon dissection of their brains. The *Ptbp2*-EmxKO cortex appeared relatively normal at birth. However, while the control cortices thicken and grow in size during the first three postnatal weeks, largely due to development of the neuropil and white matter, the EmxKO cortex displayed widespread neuronal death and degeneration (*Figure 3C–E*). Brain structures not subject to PTBP2 loss, such as striatum, developed normally. The knockout tissue showed a progressive loss of structural integrity making it difficult to section and stain. Comparing coronal sections of mutant and control brains at P5, the

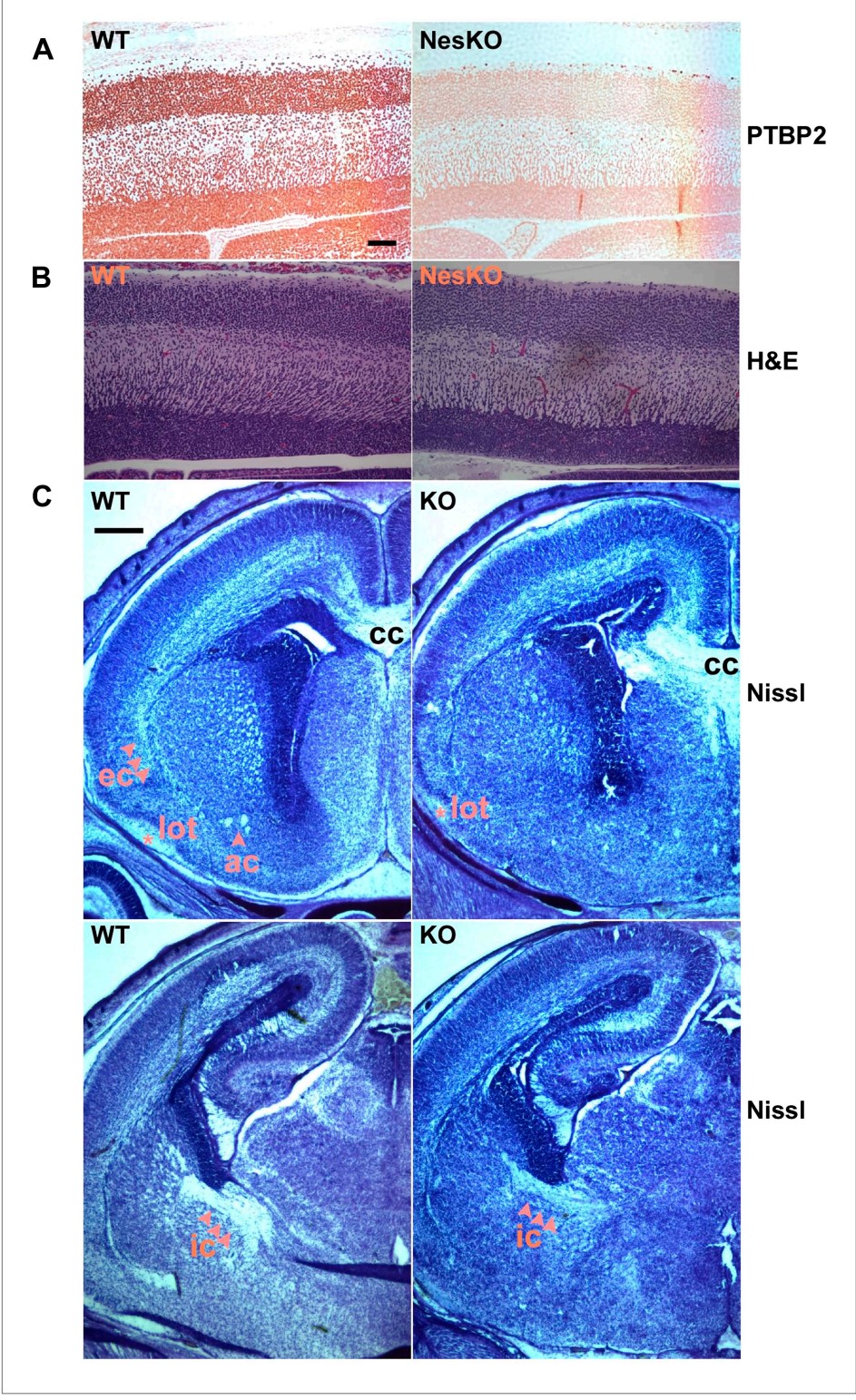

**Figure 2**. Embryonic brain development appears largely normal in the absence of PTBP2. (**A** and **B**) Sagittal sections of E18.5 cortex from control and NesKO mice. (**A**) Staining with anti-PTBP2 antibody, control cortex shows PTBP2 expression in all neuronal nuclei, whereas NesKO cortex has lost PTBP2 expression in the vast majority of neurons. (**B**) H&E staining showing that NesKO cortex has similar thickness and largely normal morphology. Scale bar = 100 µM. (**C**) Nissl stained coronal sections of E18.5 cortex of wild-type and NesKO brain at two

*Figure 2. Continued on next page*

*Figure 2. Continued*

rostral–caudal planes. In the knockout brain, several major axonal tracks including the lateral olfactory tract (lot), internal capsule (ic) and external capsule (ec) were reduced, and the anterior commisure (ac) was missing. Corpus collosum, the major axon bundle connecting the two hemispheres, was present in the knockout brain. Scale bar = 250 µM. Staining was done on at least four knockout mice and four wild-type littermate controls from two litters.

EmxKO cortex was substantially thinner than control cortices. Nissl staining revealed widespread pyknotic nuclei within the EmxKO cortical plate as a result of neuronal cell death (*Figure 3D*). Further staining with more specific cell death markers showed evidence of both necrotic and apoptotic cell death but was difficult to quantify in the degenerating tissue (data not shown). Understanding the mechanisms initiating cell death in these tissues will require additional analyses. By P11, when control cortex had substantially thickened with well-defined cellular layers, the EmxKO cortex had further degenerated with additional loss of tissue integrity. The lateral ventricles were enlarged in EmxKO brains at both P5 and P11, and the corpus callosum was missing (*Figure 3E* and data not shown). Other brain structures where *Ptbp2* was inactivated were also defective in the EmxKO; the mutant hippocampus was significantly reduced in size and the lamination in the olfactory bulb was highly disrupted, with the Mitral cell layer entirely absent (data not shown).

The postnatal lethality of *Ptbp2*-EmxKO mice demonstrates a continued requirement for PTBP2 to allow neuronal maturation and survival in the developing postnatal cortex. The survival of these mice past birth suggests that the neonatal lethality of *Ptbp2* null and NesKO mice is due to loss of PTBP2 in early maturing regions of the CNS critical for basic physiological function in the neonate.

While initial neurogenesis, as well as patterning and specification of the brain, appeared largely normal in the absence of PTBP2, the phenotype of the EmxKO mice indicated a failure of later neuronal development. The paralysis phenotype of *Ptbp2* pan-neural knockout pups also implied defects in synaptic transmission. To assess whether neural development proceeded to later stages, such as synapse formation, in the absence of PTBP2, we examined the expression of several important synaptic proteins in the brains of E18 *Ptbp2* null mice. The mutant brains exhibited strikingly lower levels of multiple synaptic markers (*Figure 4A*), including PSD95, the neurotransmitter receptor NR2B, the major synaptic vesicle protein Synaptophysin (SYP), and the trans-synaptic scaffolding protein Neuroligin1 (NLG1). Other synaptic proteins, such as the calcium/calmodulin-dependent protein kinase Cask and the Glutamate Receptor GluR2 also declined. These results indicate that cells may be arrested or delayed in their maturation or that synapse formation may be otherwise defective in the absence of PTBP2.

Interestingly, heterozygous *Ptbp2*$^{+/-}$ mouse brains express the PTBP2 protein at about half normal levels (*Figure 1F*). The effect of this partial depletion on synaptic protein expression and on splicing of target transcripts was variable (*Figure 4A*, and see below). For some targets, protein levels in the heterozygous brains were intermediate between the wild type and homozygous knockout (GluR2 and NR2B, *Figure 4A*, also see CamKinase 2b below). In other cases, the heterozygote appeared similar to the wild-type mice, expressing close to normal levels of PSD95, Synaptophysin, Neuroligin1, and Cask. Given that *Ptbp2* heterozygous mice are viable, a single copy of *Ptbp2* is apparently sufficient to support development. It will be interesting to examine the behavior and physiology of these heterozygous mice.

To investigate how developing neurons are affected by the loss of PTBP2, we cultured neurons from knockout and wild-type embryos. Dissociated mouse neurons cultured in vitro progress through several stages of maturation that parallel their in vivo development (*Dotti et al., 1988*; *Palmer et al., 1997*; *Arimura and Kaibuchi, 2007*). These include the initial extension of multiple immature neurites (days in vitro, DIV1-2), establishment of axon–dendrite polarity (DIV2-4), axon and dendrite outgrowth (DIV4-15), and finally synapse formation and maturation (after DIV10). Immediately after dissociation and plating, cortical neurons from E15-16 *Ptbp2*-NesKO mice exhibited similar viability to control neurons. The *Ptbp2*-KO neurons grew extensive neurites during the first week in culture and were indistinguishable from control cultures when stained with the dendritic marker MAP2 (*Figure 4B*). However during the second week, increased signs of neuronal death appear in cultures of *Ptbp2*-KO neurons, including neurite retraction, membrane blebbing and shrinkage, and nuclear condensation. All neurons lacking PTBP2 die by the end of third week of culture. A similar catastrophic cell death occurred during the third week of culture in hippocampal neurons cultured from E18 *Ptbp2* null mice (*Figure 4C*). The timing and scale of the neuronal death in culture is similar to the wide-spread

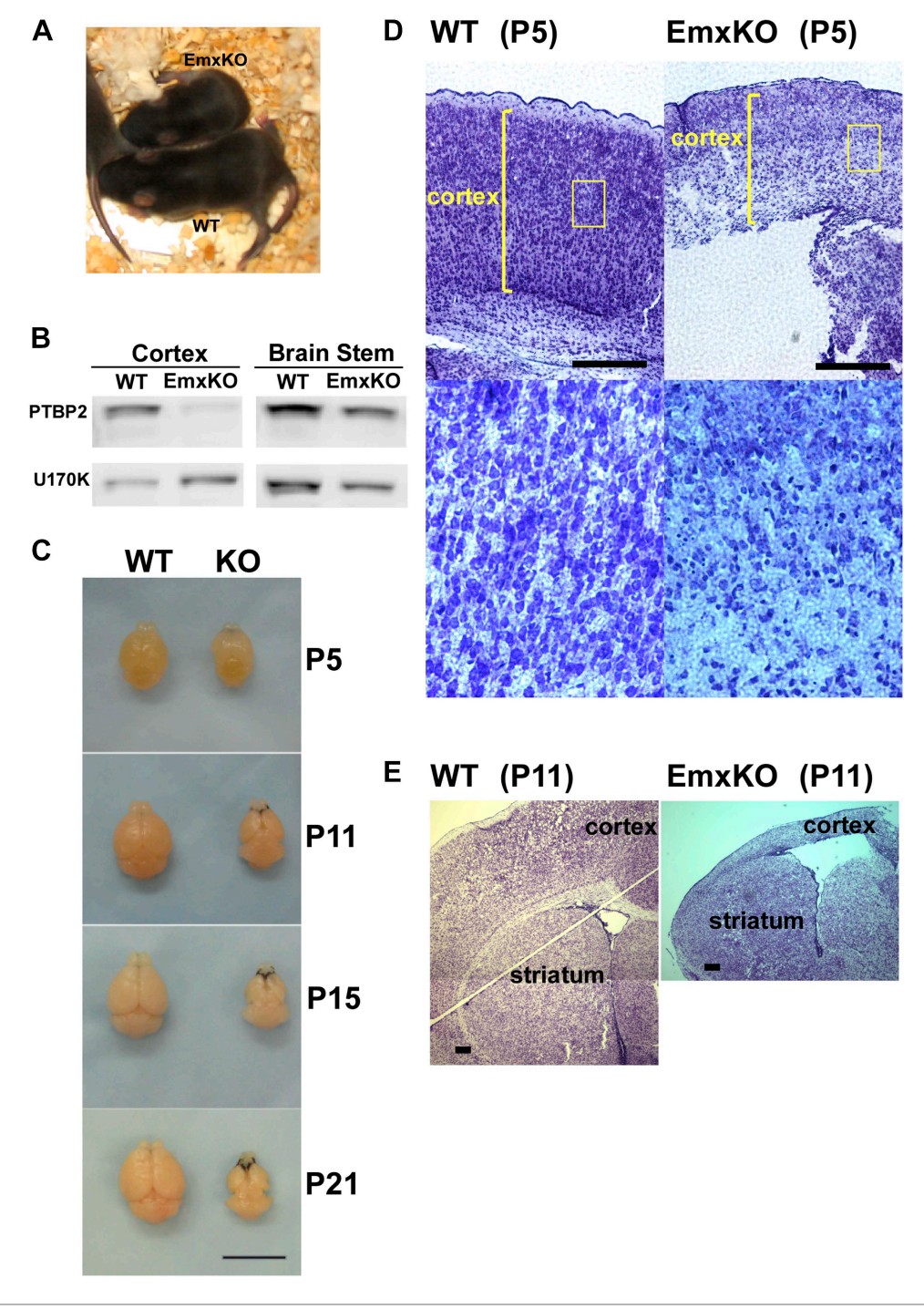

**Figure 3**. PTBP2 is required for postnatal cortical development. (**A**) *Ptbp2* EmxKO mutants display slow growth (shown at P11) and die around weaning. (**B**) Immunoblot for PTBP2 at P21 confirms that its expression is largely eliminated in cortex, but remains unchanged in other structures such as brain stem. (**C**) Postnatal development of the cortex was disrupted in EmxKO mice. Mutant cortical tissue failed to thicken as in wild type and degenerated. Scale bar = 1 cm. (**D**) Nissl stained coronal sections of control and EmxKO brain at P5 showing loss of cell density and degeneration. Enlarged panels show a loss of nuclei in the EmxKO tissue. (**E**) Coronal sections at P11, similar to (**D**). Scale bars in **D** and **E** indicate 0.5 mm. Staining was done on at least four knockout mice and four wild-type littermate controls from two litters.

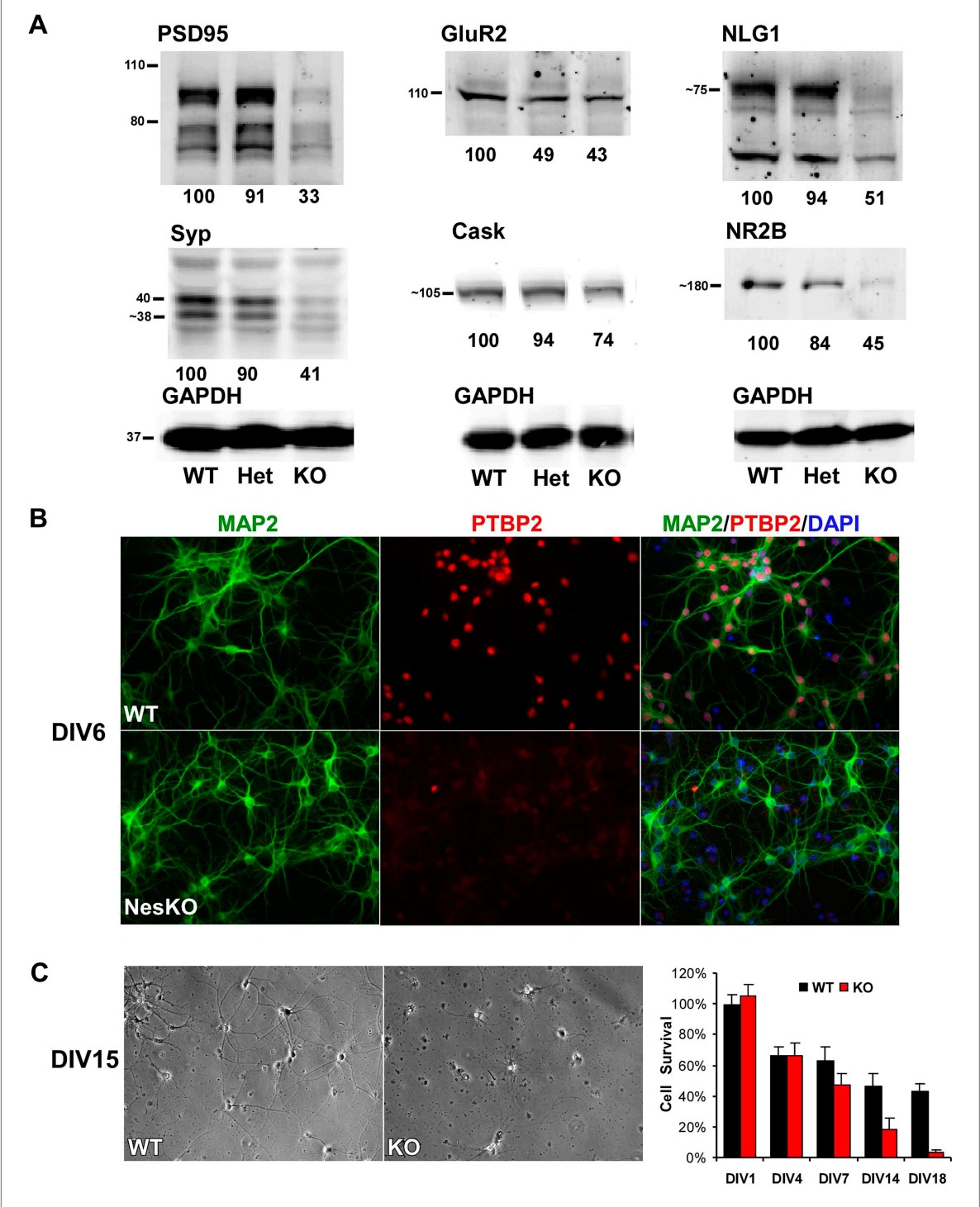

**Figure 4**. Loss of PTBP2 leads to reduced synaptic protein expression in vivo and to early cell death in primary cell culture. (**A**) Immunoblots of whole brain lysates from wild type, heterozygous (+/−), and knockout (−/−) mouse at E18.5 show substantial reduction of synaptic proteins in knockout brain. (**B**) Dissociated cortical cultures (Days in vitro, DIV6) of wild-type and *Ptbp2*-knockout neurons stained for PTBP2 and the neurite marker MAP2. Neurons lacking PTBP2 survive plating and short-term culture with similar efficiency to wild type, and extend multiple MAP2-positive neurites. (**C**) PTBP2-deficient hippocampal cultures show increasing neuronal death starting from second week of culture and do not survive past 3 weeks. A phase contrast image of cells at DIV15 is shown, with the quantification of cell numbers plotted to the right. Starting viable cell numbers averaged 417 cells from wt and 442 cells from knockout based on cultures from four embryos of each genotype at DIV1. Percent Survival is the fraction of live cells at each time point relative to the wild type average at DIV1. Error bars are standard deviations derived from four cultures of each genotype at each time point.

degeneration seen in the postnatal (P5-P11) Emx-KO brains. Both in vivo and in vitro, PTBP2 is clearly required for the survival of forebrain neurons.

## *Ptbp2* knockout brain exhibits extensive changes in alternative splicing

Splicing regulators generally have many target transcripts and it is expected that the dramatic phenotype of PTBP2 loss will be derived from multiple deficits. To assess transcriptome changes caused by the loss of PTBP2, we carried out genomewide expression and splicing analyses of the nesKO mice, initially using splicing-sensitive exon junction microarrays (MJAY) analyzed by Omniviewer, and later using RNAseq and SpliceTrap analysis (*Srinivasan et al., 2005*; *Wu et al., 2011*; *Shen et al., 2012*). These analyses identified a large set of cassette exons and other alternative splicing events that are mis-regulated in developing (E18) *Ptbp2*⁻/⁻ brains (Supplementary files 1–3 in Dryad [*Li et al., 2014*]). Alternative splicing events that changed between wild type and mutant brains were ranked by Sepscore and q-value for microarray analysis, and by delta PSI (percent spliced in) and false discovery rate (FDR) for RNAseq ('Materials and methods'). Using cutoffs of a Sepscore of 0.345 for the arrays and a 10% change in PSI for the RNAseq, we identified 359 PTBP2 target cassette exons by array (Supplementary file 2 in Dryad [*Li et al., 2014*]) and 597 exons by RNAseq (Supplementary file 3 in Dryad [*Li et al., 2014*]). These included 92 exons found by both methods. Exons identified by RNAseq but not microarray included exons that were not represented on the array, exons exhibiting complex patterns of splicing, and a few exons where particular exon probes seemed to fail on the array. Alternative splicing events identified by array but not RNAseq included exons that did not pass the deltaPSI and/or overall expression filters, and a few exons that were absent from the SpliceTrap database. Combining the two methods provided a significant increase in the total set of splicing changes identified in the mutant mice.

Assaying additional control and mutant mice as biological replicates, a large set of splicing events was further validated by RT-PCR (77 out of 88; *Figure 5A*; Supplementary file 1 in Dryad [*Li et al., 2014*]). These included exons identified by either of the two profiling methods, as well as several exons previously identified as PTBP1 targets. Both the array and RNAseq methods yielded validation rates above 70%. The majority of identified cassette exons (379 of 597 from the RNAseq analysis) showed increased splicing in the absence of PTBP2 (Supplementary file 3 in Dryad [*Li et al., 2014*]), suggesting that PTBP2 functions more often as a splicing repressor than as a splicing activator, similar to the observations of PTBP1 (*Keppetipola et al., 2012*).

Altered splicing was observed in many known direct PTBP1/PTBP2 targets including PSD95 (*Dlg4*), *Atp2b1*, *Bin1*, *Dst* and *Fn1* (*Boutz et al., 2007*; *Makeyev et al., 2007*; *Zheng et al., 2012*). Direct targeting by PTBP2 can also be inferred from crosslinking-immunoprecipitation (CLIPseq) data (*Licatalosi et al., 2012*; *Ule et al., 2003*). For exons whose percent-spliced-in value (PSI) changed by more than 20% in the knockout, approximately 70% are reported to have significant PTBP2 CLIP clusters in their adjacent introns in E18 mouse brain (*Licatalosi et al., 2012*) (SZ unpublished data). Exons in *Agrn*, *Arhgef7*, *Dbn1*, *Erc1* and *Sorbs2*, whose splicing is altered in the PTBP2⁻/⁻ brain and which are also expressed in HeLa cells, were also identified as PTBP1 targets in a HeLa CLIP analysis (*Xue et al., 2009*). These data indicate that the majority of isoform changes identified in the knockout are likely to be directly regulated by PTBP2. Nevertheless, it should be noted that the analyzed brains developed to E18 in the absence of PTBP2 and it is expected that some splicing changes will be indirect effects of other changes in the developing mice.

Validated PTBP2-dependent splicing events include many transcripts with important neuronal functions, including the postsynaptic scaffolding proteins PSD95 (*Dlg4*) and gephryn (*Gphn*), the neurotransmitter receptor GABA$_A$ Rγ2 (*Gabg2*), and the key calcium signaling molecules CamK2a and CamK2b (*Figure 5AB*, *Figure 5—figure supplement 1*; *Li et al., 2014*). As a general assessment of the PTBP2 regulatory network, gene ontology analyses were performed on the set of PTBP2-dependent splicing changes relative to the set of transcripts identified by RNAseq as expressed in mouse brain. Biological Process annotations that were significantly enriched in the PTBP2 target set (p<0.001) included 'regulation of transcription', 'synaptic transmission', 'transmission of nerve impulse', 'synapse organization', 'cell morphogenesis', 'endocytosis', and 'neuron projection development' (Supplementary file 4 in Dryad [*Li et al., 2014*]). All of these functions are dynamically changing during the period of neuronal maturation when PTBP2-dependent defects are observed.

We also analyzed the RNA-seq data using CuffDiff to identify changes in overall transcript expression (rather than splicing) in the *Ptbp2* mutant brain (Supplementary file 5 in Dryad [*Li et al., 2014*]). Some of these transcripts are likely to be reduced through direct PTBP2 effects on post-transcriptional processes. These include five transcripts containing PTBP2-dependent exons with mapped CLIP clusters whose

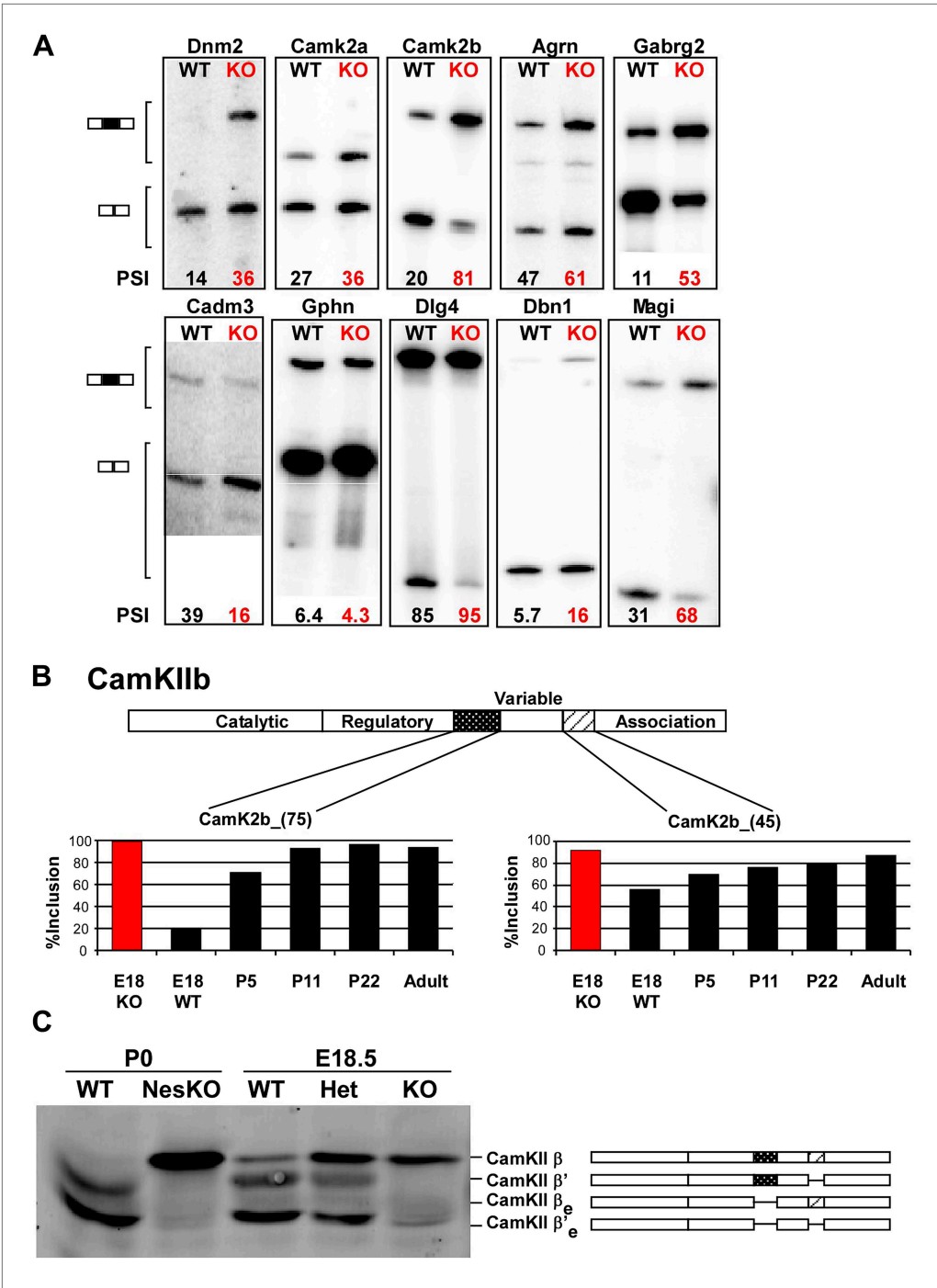

**Figure 5**. PTBP2 regulates a large set of splicing targets important to neuronal development and function. (**A**) Sample RT-PCR gels showing aberrant splicing ratios for selected targets important for neuronal development and function. (**B**) Two alternative exons in *Camk2b* are developmentally regulated. In plots of RT/PCR measurements, normal mouse brain shows low inclusion of these exons in the embryo (E18) and a postnatal increase (P5, P11 and P22) in splicing to near full inclusion in adult brain. In the PTBP2 null brain these exons exhibit adult levels of splicing at E18. (**C**) The premature switch in CAMK2B protein isoforms is observed by immunoblot in both *Ptbp2* null (KO) and NesKO mice (E18.5-P0).

The following figure supplements are available for figure 5:

**Figure supplement 1**. Quantification of RT/PCR analysis for multiple exons mis-regulated in *Ptbp2* Nes-KO brain.

change in splicing is predicted to lead to nonsense mediated decay. Other changes in transcript expression are likely to be an indirect result of developmental defects associated with the loss of PTBP2. Gene ontology analyses were performed on genes exhibiting greater than a twofold change in the knockout brain (Supplementary files 6 and 7 in Dryad [*Li et al., 2014*]). Relative to all transcripts expressed in wild-type brains, upregulated transcripts were highly enriched for GO terms associated with innate immunity and the inflammatory response (p<0.001). Given the phenotype of the mice, these functional enrichments likely result from the degeneration seen in the *Ptbp2* knockout brains, rather than direct PTBP2 targeting of the transcripts. Down-regulated transcripts were only weakly enriched for particular bioprocess terms. Only terms related to cell adhesion had enrichment p values below 0.01. Terms related to axonogenesis and axon choice point recognition were less significantly enriched among the down-regulated transcripts.

## PTBP2 controls a splicing regulatory program specific to early neuronal maturation

We find that many of the identified PTBP2 targets display a common pattern of temporal regulation in the developing wild-type brain. These transcripts contain exons that are mostly excluded in the embryonic brain and whose splicing increases during the first 3 weeks after birth (*Figure 5B*, *Figure 5—figure supplement 1* in Dryad [*Li et al., 2014*]). For much of the brain, this time period represents a key phase of neuronal maturation when neurons project processes and establish synapses. The normal shift in splicing of these transcripts coincides with the known postnatal decrease in PTBP2 levels in the brain (*Zheng et al., 2012*), and the mutation of *Ptbp2* leads to a premature switch from embryonic to adult isoforms. In one example, the *Camk2b* gene contains two alternative exons that are mostly excluded from embryonic *Camk2b* transcripts, but are included in the adult isoform (*Brocke et al., 1995*). These two cassette exons insert short peptide sequences into the variable domain of the protein to alter its autophosphorylation, its response to $Ca^{2+}$ oscillations, and its association with F-actin (*Figure 5B*) (*Brocke et al., 1999*; *Bayer et al., 2002*; *O'Leary et al., 2006*). In the *Ptbp2* mutant brains, the two adult exons are nearly completely included by embryonic day 18 (*Figure 5B*). These Camk2b protein isoforms can also be distinguished in immunoblots that demonstrate a dramatic switch from the embryonic (β', βe and β'e) proteins to the adult (β) isoform (*Figure 5C*).

Multiple proteins important to neuronal differentiation and synaptic function show a precocious switch to adult isoforms in the *Ptbp2* knockout mice (*Figure 5—figure supplement 1*). Like CamK2b, the adult and embryonic isoforms of the postsynaptic protein Drebrin (Dbn1) and the neural cell adhesion molecule NCAM have been shown to be functionally different (*Polo-Parada et al., 2005*; *Hata et al., 2007*; *Mizui et al., 2009*; *Kojima et al., 2010*). For these genes, adult RNA and protein products are both expressed much earlier in the *Ptbp2* mutant than in wild-type mice (*Figure 5A* and data not shown). Other key proteins targeted by PTBP2 include dynamin1 (Dnm1), an essential protein in synaptic vesicle endocytosis and trafficking, Magi1, a scaffolding protein at the postsynaptic density, the potassium channel Kcnq2, a critical regulator of neuronal excitability, and calcineurin A (*Ppp3ca*), an important phosphatase affecting calcium signaling (*Figure 6A*, *Figure 5—figure supplement 1*). The adult isoforms of these proteins are presumably needed in mature synaptically active neurons.

Adult brain exhibits many splicing changes from the embryo that result from changes in a wide variety of regulators. We find that within a set of 1143 exons whose splicing increases between the embryo and adult, 17% are derepressed by the loss of PTBP2 (AH unpublished observations). Thus, the PTBP2 regulatory program accounts for a substantial fraction of the regulated splicing events controlled during brain development. The defective development of the *Ptbp2*$^{-/-}$ brain indicates that early switching to these adult isoforms is deleterious to neurons as they mature.

RNA isolated from embryonic brain at E18 will be derived from a variety of structures at different stages of development. To examine splicing within a more defined population of cells coincident with the normal downregulation of PTBP2, we isolated RNA from the EmxKO cortex for RNAseq at P1. These analyses identified splicing changes in many of the targets previously identified in the NesKO mice (Supplementary file 8 in Dryad [*Li et al., 2014*]). As seen in the NesKO at E18, the mutually exclusive exons 9a and 9b of the dynamin1 (*Dnm1*) transcript also display aberrant splicing in the EmxKO cortex (*Figures 6A and 7A*). The adult exon 9a was spliced at significantly higher levels (79%) in the EmxKO than in control mice (56%). The embryonic and adult isoforms of dynamin1 are functionally distinct and their activity has been investigated in mutant mice, where a missense mutation in exon 9a results in impaired synaptic function, seizures and other behavioral defects (*Boumil et al., 2010*). Thus, PTBP2 is playing a key role in the regulation in this postnatal switch in Dnm1 isoforms.

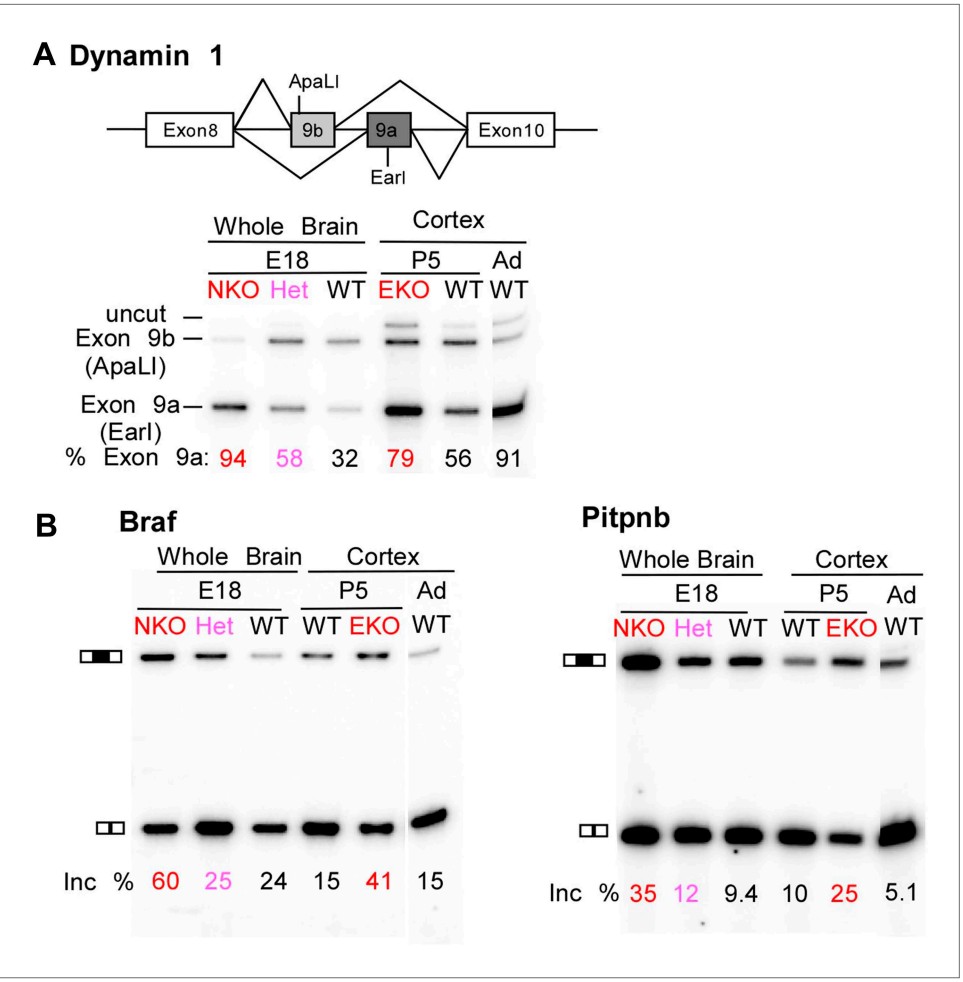

**Figure 6**. Premature and aberrantly high splicing of PTBP2 target exons in E18 Nes-KO whole brain and P5 Emx-KO cortex. (**A**) RT/PCR of Dynamin1. Dynamin1 contains a pair of mutually exclusive exons 9a and 9b that normally switch from 9b to 9a during development. In both the Nes-KO and Emx-KO there is a premature switch to the 9a isoform. Note that the heterozygous Nes-KO exhibits an intermediate level of exon 9a splicing. The high level of 9a splicing can be seen in the wild-type adult cortex (Ad WT). (**B**) RT/PCR of Braf (left) and Pitpnb (right). Braf and Pitpnb each contain a PTBP2 repressed exon that is spliced at aberrantly high levels in the *Ptbp2* KO mice. Braf exon 9 and Pitpnb exon 11 are spliced into the mRNAs two to three times more frequently in the embryonic (E18) or postnatal (P5) mutant mice (NKO and EKO), and at higher levels than normally seen in the wild-type adult (Ad WT). The percent exon inclusion is shown at the bottom.

Several exons were found to be included in the *Ptbp2* knockout brain at levels much higher than those observed at any point of normal development. These include an exon in B-raf (Braf), a protein kinase in the Ras-Raf-MEK-ERK signaling pathway (*Maurer et al., 2011*). This exon is normally included in no more than 25% of the total Braf transcript. In contrast, 60% of Braf mRNA in E18 null brain, and 41% of the mRNA in P5 EmxKO cortex contains the exon (*Figure 6B*). The PITPbeta (phosphatidylinositol transfer protein beta, *Pitpnb*) transcript, involved in inter-organelle lipid transfer and inositol signaling, contains an exon that can be skipped or included to produce alternative C-termini of the final protein (*Cockcroft, 2001*). In wild-type mouse brain, this exon is included in less than 10% of the Pitpnb mRNA at all points of normal development. However, in the absence of PTBP2 splicing of this exon increases to more than 25% (*Figure 6B*). These and other exons appear to be especially sensitive to loss of PTBP2 (*Figure 5—figure supplement 1*). Although reduced in expression after differentiation, PTBP2 continues to be present at moderate levels into adulthood. This remaining PTBP2 may serve to maintain the partial repression of some exons. In addition to the precocious expression of adult isoforms, the deleterious effect of PTBP2 loss may also result from the aberrantly high expression of isoforms normally present at lower levels.

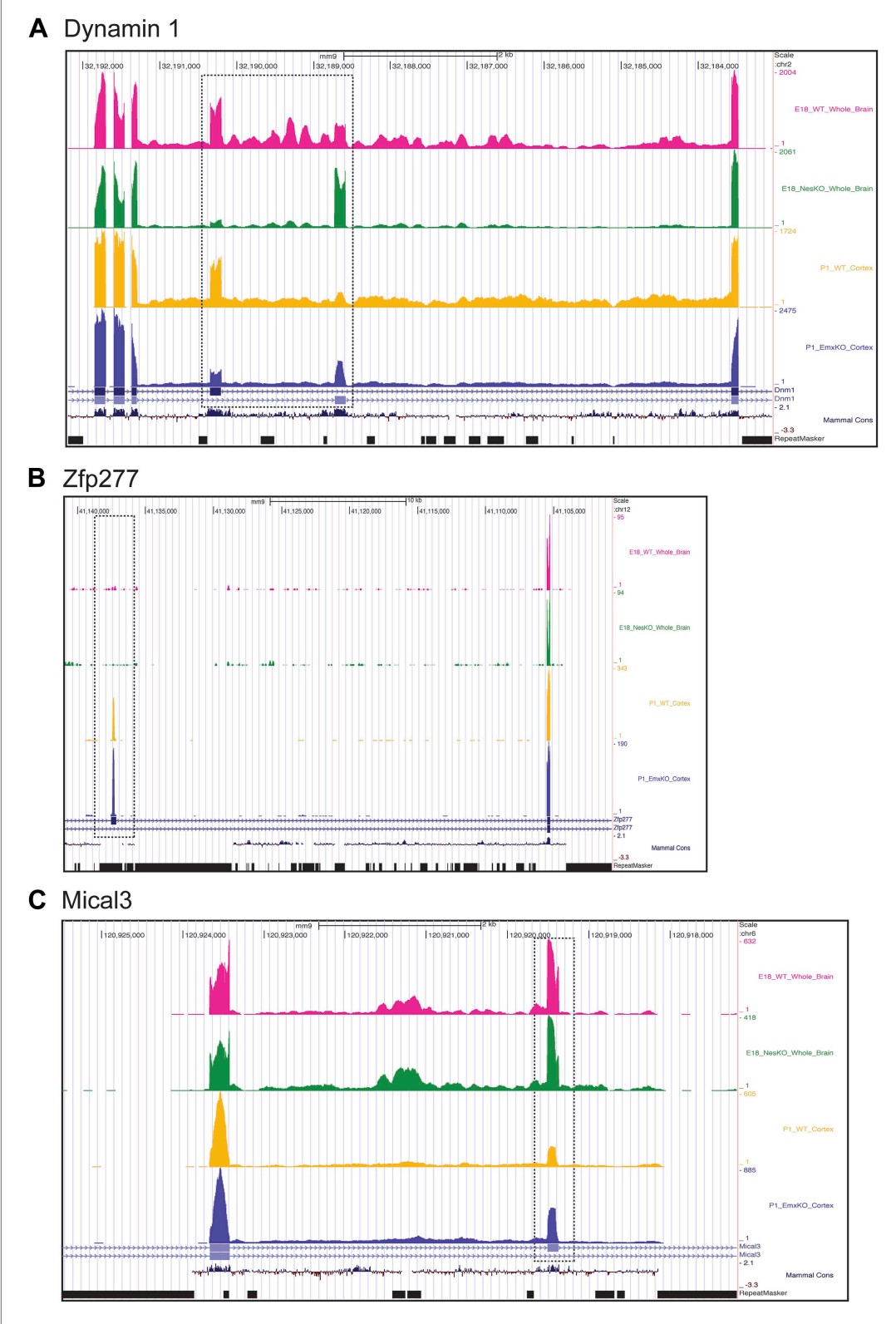

**Figure 7**. Exons can respond to the loss of PTBP2 differently in E18 whole brain or P1 cortex. Genome browser tracks of aligned RNAseq reads from E18 whole brain, NesKO (green) and wild type (pink), and from P1 cortex, EmxKO (blue) and wild type (yellow). (**A**) For Dynamin 1, exons 9a and 9b are boxed. The two knockout samples show a different degree of shift towards the use of downstream exon 9a. (**B**) An exon in the Zinc finger protein Zfp277
*Figure 7. Continued on next page*

*Figure 7. Continued*

(boxed) is entirely excluded in E18 brain. In postnatal cortex this exon has been strongly induced, but is still regulated by PTBP2 as indicated by the stronger induction of the exon in the Emx-KO. (**C**) An exon in the microtubule associated monooxygenase Mical3 is fully spliced at E18, regardless of the presence of PTBP2. In postnatal cortex this exon becomes repressed and this repression is in part dependent on PTBP2.

In addition to splicing changes seen in the NesKO whole brain, new PTBP2-dependent splicing events were identified in EmxKO cortex (*Figure 7BC*, Supplementary file 8 in Dryad [*Li et al., 2014*]). The zinc finger protein transcript *Zfp277* contains two exons whose splicing increases substantially in postnatal cortex compared to E18 whole brain. Interestingly, one of these exons shows greater induction in the Emx-cre KO than in wildtype, indicating that it is partially repressed by PTBP2 in this tissue. In contrast, this exon remains fully repressed in E18 brain even in the absence of PTBP2 (*Figure 7B*). Conversely, several exons in the *Mical3* transcript are repressed in P1 cortex but not E18 whole brain. One of these exons loses this cortical repression in the absence of PTBP2, even though it is unaffected by PTBP2 depletion at E18 (*Figure 7C*). These exons showing PTBP2 dependence in P1 cortex but not E18 whole brain indicate that the PTBP2 target transcripts are subject to other splicing regulatory programs that are also changing during development, with exons exhibiting different regulatory factor dependencies at different developmental stages.

The many adult isoforms misregulated in the *Ptbp2*$^{-/-}$ embryos greatly expands earlier observations of exon 18 of the *Dlg4* (PSD95) transcript, whose skipping generates a non-productive transcript subject to nonsense-mediated decay. The splicing of this exon is required for PSD95 protein expression, and its repression by PTBP2 keeps PSD95 levels low until late in neuronal maturation (*Zheng et al., 2012*). We now find that this exon is part of a large program of coordinated splicing changes. This PTBP2-driven splicing transition takes place subsequent to the earlier splicing switch driven by PTBP1 depletion as neurons are born (*Figure 8*).

## Discussion

We find that the splicing regulator PTBP2 is required for pre and postnatal brain development. The loss of PTBP2 in the complete CNS or in particular neuronal lineages does not greatly affect lineage commitment or developmental patterning. However, postmitotic neuronal maturation and survival are severely impaired. Mice with Emx-cre driven depletion of PTBP2 from developing cortex survive through about 3 weeks of postnatal development during which time the cortex degenerates. Neuronal cell death is also seen in *Ptbp2* knockout neurons in culture. These cells initiate differentiation from progenitor cells and progress through the first stages of post-mitotic differentiation, with the extension of neuritic processes and expression of early neuronal markers. However, they subsequently undergo a catastrophic failure in the second to third week of culture and die prior to synaptogenesis. Thus, the PTBP2-driven splicing program plays a key role in the maturation and survival of neurons.

In keeping with the neuronal maturation defect, we find PTBP2 target exons in many transcripts affecting neurite outgrowth, axon guidance, synaptic assembly, and synaptic function. In the mutant embryos, a large number of genes exhibit early expression of spliced isoforms that are normally expressed only in the adult brain. For these genes, the presence of PTBP2 in developing cells serves to prevent expression of a functionally distinct adult isoform until late in neuronal maturation. This pattern of regulation is seen for NCAM, Cam Kinase II, drebrin, dynamin1, Magi1, Kcnq2, Calcineurin and the previously identified PTBP2 target PSD95 (*Dlg4*). These proteins all play key roles in the development and function of the neuron including both its pre and postsynaptic specializations.

PTBP2 is present at low levels in nestin-positive neuronal progenitor cells. When these cells exit mitosis and begin differentiation, PTBP1 expression is repressed and PTBP2 expression is induced (*Figure 1A*; *Boutz et al., 2007*). PTBP2 levels remain high in the differentiating cells and then drop late in maturation, with adult neurons expressing moderate levels. It was previously shown that some exons are more strongly affected by PTBP1 than by PTBP2, whereas for other exons the regulatory effect of the two proteins is nearly equal (*Boutz et al., 2007*; *Makeyev et al., 2007*; *Llorian et al., 2010*; *Tang et al., 2011*; *Keppetipola et al., 2012*). The expression profile of the PTB proteins thus creates three regulatory states during neuronal differentiation (*Figure 8*). PTBP1 maintains repression of many exons in neuronal progenitor cells. As cells begin to differentiate and shift to PTBP2 expression, exons that are more sensitive to PTBP1 begin to be spliced. These include the *c-src* N1 exon and exon 8A from the *Cacna1c* calcium channel transcript (*Markovtsov et al., 2000*; *Tang et al., 2011*). Later in neuronal

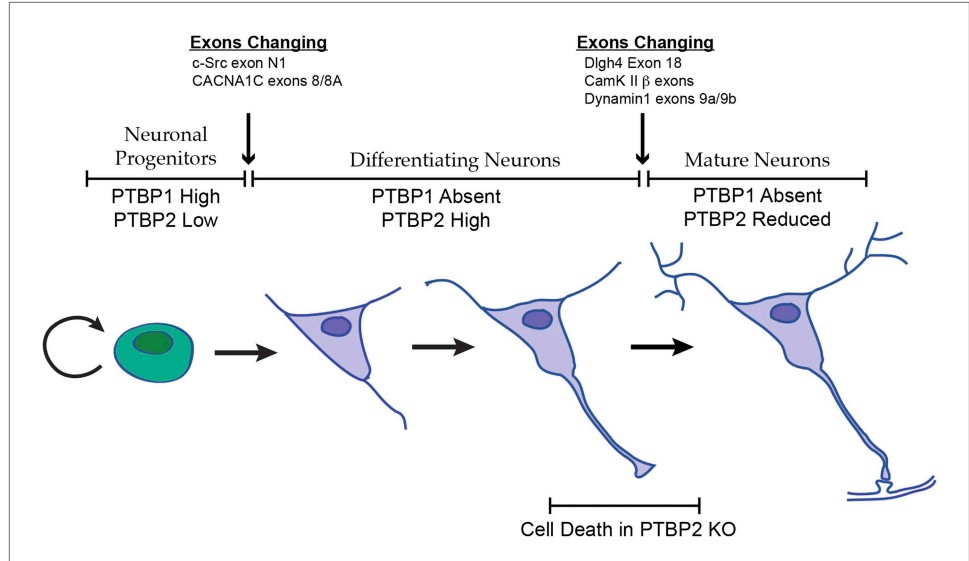

**Figure 8**. Changes in the expression of the two PTB proteins define three splicing regulatory states during neuronal differentiation. Neuronal progenitor cells primarily express PTBP1. When these cells are induced to differentiate, PTBP1 is repressed and PTBP2 is induced. This switch in RNA binding proteins causes changes in the splicing of exons that are more sensitive to PTBP1 such as exon 8 of the *Cacna1c* calcium channel transcript. Other exons that are sensitive to both proteins maintain their repression during the early stages of neuron differentiation, when PTBP2 is high. When these cells finally mature and form synapses, PTBP2 is downregulated. This leads to changes in a splicing regulatory program that includes exons in Cam Kinase 2 and Dynamin 1. These exons are found in mRNA isoforms associated with adult brain and are precociously expressed in PTBP2 knockout brains, leading to cell death prior to final maturation.

maturation when PTBP2 levels drop, a second transition occurs with changes in the splicing of a new group of transcripts. We previously found that PSD95 exon 18 is controlled during this second transition (*Zheng et al., 2012*). We now identify a large network of exon targets that have this common regulatory profile and are precociously spliced in the *Ptbp2⁻/⁻* mice. Thus, one function of the PTBP2 protein is to maintain repression of a subset of PTBP1 target exons after PTBP1 is depleted in early differentiation. The differing sensitivity of target exons either to just PTBP1 or to both PTBP1 and PTBP2 presents an interesting mechanistic question of how an exon can be structured to respond to one protein or to both.

A recent paper described the phenotype of a full germline *Ptbp2* null mouse (*Licatalosi et al., 2012*). As we observe with both germline and pan-CNS conditional knockouts, this mouse exhibited neonatal lethality, precluding examination of later neuronal maturation. Perhaps because of this neonatal lethality, the authors focused on PTBP2 expression in neuronal progenitor cells. Although we also see low expression in these cells (*Figure 1A*), we find the primary period of PTBP2 expression to be during neuronal maturation (*Boutz et al., 2007*; *Tang et al., 2011*; *Zheng et al., 2012*). This study reported sporadic observation of cortical progenitors undergoing ectopic mitoses away from the apical edge of the ventricular zone in *Ptbp2* null mice (*Licatalosi et al., 2012*). This defect was in keeping with an observed change in *Numb* pre-mRNA splicing. We have not observed ectopic mitoses in either the germline null or the pan-CNS conditional null mice, and found only minor changes in *Numb* splicing. However, this phenotype was reported to be sporadic and a careful comparison of the phenotypes from the different mutations will require matching their genetic backgrounds. Licatalosi et al. also report microarray identification of PTBP2 target exons in germline null brain that largely overlap with exons we identify in the pan-CNS knockout brain by microarray and by RNAseq. They suggest a role for the PTBP2 splicing program in neurogenesis. We do not see defects in neurogenesis and commitment to neuronal lineages, but find a dramatic defect in the later maturation of neurons after their birth. This is in keeping with both the developmental timing of PTBP2 expression and the precocious expression of adult isoforms observed by both groups in the knockout mice.

Interestingly, the later Emx-knockout mice show some differences from the NesKO in their ensemble of PTB2 target exons. These transcripts may be subject to regulatory events that occur after birth and are thus not observable in mice subject to neonatal lethality. Exons observed to change in the EmxKO but not in the NesKO mice point to the dynamic nature of splicing regulation during neuronal

development. Most exons are controlled by combinations of factors, and there is likely to be extensive intersection between the PTBP2 regulatory program and splicing programs driven by other splicing factors (*Calarco et al., 2011*). In the future, it will be interesting to focus on the development of particular circuits and assess the temporal expression profiles of multiple other splicing regulators. Such studies will allow examination of how the different splicing regulatory networks are integrated.

Our analysis has focused on the dramatic changes in alternative splicing caused by loss of PTBP2. PTBP1 is known to also affect miRNA function and cytoplasmic translation (*Sawicka et al., 2008*; *Fred and Welsh, 2009*; *Gorospe et al., 2011*; *Kafasla et al., 2012*; *Xue et al., 2013*). PTBP2 will likely also serve similar roles, and a portion of the *Ptbp2* null phenotype may also derive from the loss of these other functions. In the CAD neuronal cell line, PTBP1 has been reported to cause retention of selected introns in transcripts for several important synaptic proteins, leading to their downregulation through nuclear RNA decay (*Yap et al., 2012*; *Yap and Makeyev, 2013*). These introns appear to be efficiently spliced in embryonic brain, as might be expected after the downregulation of PTBP1. PTBP2 may have similar activity on a different set of target transcripts. So far we have not seen a clear example of this, but one possibility is *Ngrn*, which contains a 3′-terminal intron that is spliced more efficiently in the Emx-KO than in wild type (data not shown).

As seen with other splicing regulators, the *Ptbp2* mutation is highly pleiotropic. Future studies will examine the functional roles of individual PTBP2 target transcripts. We are particularly interested in two processes. The establishment of axonal/dendritic polarity that occurs early in differentiation may coincide with the loss of PTBP1 protein and gain of PTBP2. In contrast, synaptic assembly occurs late in maturation and requires the downregulation of PTBP2. Many identified synaptic proteins change their splicing at this stage and it will be very interesting to examine how their isoform switching affects synaptic architecture and function.

## Materials and methods

### Generation of the *Ptbp2* conditional KO mouse

The *Ptbp2* conditional null allele was generated using homologous recombination in embryonic stem (ES) cells. The pFlox-PGK-Neo vector, containing a neomycin resistance gene (Neo) flanked by FRT and loxP sites and a diphtheria toxin gene cassette, was used for *Ptbp2* targeting. The 5′ and 3′ knockout arms of the targeting construct were generated by high-fidelity PCR amplification with Pfu polymerase of 129S2 genomic DNA. The targeting vector was linearized with Xba I and electroporated into 129S2-derived ES cells. Isolated ES cell clones were analyzed for homologous recombination. Incorporation of the 5′ and 3′ loxP sites was confirmed by Southern blotting using 5′ and 3′ probes following digestion with XbaI. Clones with the targeted *Ptbp2* allele were injected into 3.5-day C57BL/6 blastocysts, and the resulting chimeras were crossed to C57BL/6 females to achieve germline transmission of the targeted (PTBP2Neo-loxP) allele. The *Ptbp2Neo-loxP* mice were crossed to FLPe transgenic mice (129S4/SvJaeSor-Gt(ROSA)26Sortm1(FLP1) Dym/J; Jackson, Bar Harbor, ME) to remove the Neo cassette. *Ptbp2loxP* was crossed to the germline-active EIIaCre transgenic mice (gift of Stephen Smale) to obtain the *Ptbp2KO* alleles. Conditional targeting in the CNS or forebrain projection neurons was achieved in crosses with Nestin-Cre (Jackson) and Emx-Cre (gift of William Yang) mice respectively. The Cre-transgenes were all in the C57Bl/6 background. Analyzed knockouts had been backcrossed to this background for three to eight generations. At least 12 litters were generated for each Cre genotype to assess the phenotypes of each homozygous mutation. Genotyping PCR primers were TCTACTTCATTGTGTTGTTTTGT (F1), AGGCATCCATAATTACACAGTGT (F2), GATACAGCAGGCTCCCCTCA (R1), AAGTGATACAGCAGGCTCCC (R1.1), ATAAGCATTTTCTAGCACCAA (R2) and GCGCGTATCCTCAAACAAGAC (R3). 50-100 ng of purified genomic DNA was amplified by PCR using 5 μL of 2x GoTaq Green Master Mix (Promega™ M7123 with 0.5 uM of each PCR primer in a total 10 μL reaction. The F2 and R1. 1 pair yields a 302 bp product for the WT and a 413 bp product for the loxP allele. Initial denaturation was at 95°C for 2 min, followed by 30 cycles of denaturation at 95°C for 30 sec, annealing at 53°C for 30 sec and extension at 72°C for 40 sec, and final extension at 72°C for 6 min. The F2 and R3 pair yields a 709 bp product for the loxP deleted allele (KO). For these primers, initial denaturation was at 95°C for 2 min, followed by 31 cycles of denaturation at 95°C for 30 sec, annealing at 60°C for 30 sec and extension at 72°C for 40 sec, and final extension at 72°C for 6 min. The F1 and R1 primer pair yields a 184 bp product for the WT allele and 295 bp product for the loxP allele. The F1 and R2 pair yields a 455 bp product for the loxP deleted allele (KO). For these primer pairs, initial denaturation was at 94°C for 4 min, followed by 35 cycles of denaturation at 94°C for 30 sec, annealing at 55°C for 30 sec and extension at 72°C for 30 sec, and final extension at 72°C for 5 min. Animals

were housed in a 12-hr light/dark cycle with food and water available ad libitum and were maintained by the University of California, Los Angeles (UCLA) Division of Laboratory Medicine, as accredited by the Association for Assessment and Accreditation of Laboratory Animal Care. All experiments were approved by the UCLA Animal Research Committee.

## Immunoblot, immunofluorescence and immunohistochemistry

Fluorescent Immunoblots were performed on total protein from embryonic or postnatal mouse brain lysed in RIPA buffer and sonicated to homogenize samples using a W-385 Ultrasonic Processor (Heat Systems/Qsonica, Newtown, CT) at 10 Hz. After a 30 min incubation at 4°C, samples were frozen at −20°C until further processing. Lysates were diluted in 2× SDS loading buffer, heated at 95°C for 10 min, and loaded onto 10% polyacrylamide Laemmli SDS-PAGE gels. For blotting with fluorophore-conjugated secondary antibodies, transfers were performed on a Novex X-Cell mini-cell transfer apparatus (Invitrogen/Life Technologies, Grand Island, NY) onto Immobilon-FL PVDF membranes (Millipore, Billerica, MA). The membranes were blotted under standard conditions with primary antibodies overnight at 4°C, followed by washes and incubation with ECL Plex Cy5-conjugated goat α-mouse and goat α-rabbit secondary antibodies (1:2500; GE Healthcare, Pittsburgh, PA), and then scanned on a Typhoon Phosphorimager (GE Healthcare). Quantification of fluorescent signal was performed using ImageQuant 5.1 software (Molecular Dynamics/GE Healthcare, Pittsburgh, PA). The following primary antibodies were used: α-GAPDH 6C5 (Research Diagnostics, Inc., Flanders, NJ), α-PTBP2 IS2 (*Sharma et al., 2005*), α-U1 70 k, α-PSD95 and α-Synaptophysin (Millipore), CamKIIbeta (AbCam, Cambridge, UK), Drebrin (Assay Designs/ENZO, Farmingdale, NY), SV2 (Developmental Studies Hybridoma Bank, Iowa City, IA), Cask, GluR2, Synapsin, NLG1, NR2B (NeuroMab, Davis, CA).

To generate histological sections, adult mice were perfused transcardially with ice-cold PBS, followed by ice-cold 4% PFA/PBS. The brains were removed, post-fixed in 4% PFA/PBS overnight, cryoprotected in 20% sucrose-PBS, frozen in 4-methyl-butane, and stored at −80°C until use. 10-μm sections were cut on a cryostat and collected onto Superfrost plus slides (Thermo Fisher, San Jose, CA) at −80°C until use. The sections were thawed, post-fixed in 10% formalin for 10 min, rinsed twice in PBS, permeablized with 0.5% Triton in PBS for 10 min and incubated in 1% normal goat serum and 2 mg/ml BSA in PBS for 1 hr. The sections were incubated at room temperature overnight with primary antibodies, rinsed in PBS with 0.1%Triton (PBST) before incubation with Alexa-conjugated secondary antibodies for 2 hr at room temperature. Slides were rinsed in PBST and mounted with a Prolong Gold AntiFade mounting medium containing nuclear stain DAPI (Molecular Probes/Life Technologies, Grand Island, NY). In all cases, controls with no primary antibody yielded no labeling. Similar procedures were used for immunofluorescence staining of cortical precursors and cerebellar cells, except that they are fixed with 4% PFA/PBS for 20 min and incubated with primary antibody for 2 hr. Primary antibodies were used at the following concentrations: PTBP2 (*Boutz et al., 2007*), 1:200; MAP2 (Millipore) 1:500: tau-1 (clone PC1C6; Millipore) 1:200; AnkG (NeuroMab) 1:100. Secondary antibodies, Alexa488-anti-mouse IgG and Alexa568-anti-rabbit IgG (Molecular Probes) were used at 1:1000 dilutions.

## RNA isolation, semi-quantitative RT-PCR, and microarray hybridization

Mouse brain RNA was extracted in Trizol after tissue homogenization with a Tissue Tearor (Biospec, Bartlesville, OK), and quantified by absorbance (A260) using a Nanodrop-1000 spectrophotometer (Nanodrop Technologies/Thermo Fisher, San Jose, CA). Each RT reaction contained 0.5–1 μg total RNA in a 10 μl reaction with 0.25 μl Superscript II reverse transcriptase (Invitrogen). 1/10th volume of the RT reaction was subjected to PCR amplification in a 25 μl PCR reaction containing 200,000–500,000 cpm of 32P end-labeled reverse-strand primer; PCR reactions were run on an MJ Research PTC-200 thermocycler for 18–22 cycles with an annealing temperature of 55°C. A half of each PCR reaction was mixed 1:1 with 95% formamide containing 5% 10 mM Tris pH 8.0 with bromophenol blue and xylene cyanol. RT-PCR reactions were loaded onto 8% polyacrylamide, 7.5 M urea gels and electrophoresed. Gels were dried, imaged on a Typhoon Phosphorimager (GE Lifesciences), and bands were quantified using ImageQuant 5.1 software (Molecular Dynamics).

For microarray hybridization, Trizol extracted total RNA from mouse brain was further treated with DnaseI and purified by phenol/chloroform extraction and ethanol precipitation. For transcriptome-wide

splicing analysis using Affymetrix MJAY splice junction arrays, ribosomal RNAs were removed from samples using the RiboMinus Transcriptome Isolation Kit (Invitrogen) according to the manufacturer's instructions. Amplified, biotinylated cDNA samples were produced using the GeneChip Whole Transcript Sense Target Labeling and Control Reagents kit (Affymetrix, Santa Clara, CA) according to the manufacturer's instructions. RNA from three animals for each genotype (wild type and knockout) was used as biological triplicates. Labeled samples were hybridized overnight to a Mouse GeneSplice Array (Affymetrix PN 540092). Hybridized arrays were processed using the Affymetrix Fluidics Station 450 and scanned with an Affymetrix GeneChip scanner. Data were analyzed as previously described (*Srinivasan et al., 2005*).

## RNA-seq analysis

RNAseq analysis was carried out on polyA + RNA from two brains each of Nes-KO and wild-type littermates at E18, and dissected mouse cortices from Emx-KO and wild-type littermates at P1. Paired end libraries for 100 nt read length were constructed using the Illumina Tru-seq kit. The Emx-KO libraries were strand specific with the second cDNA strand eliminated with the USER enzyme (*Bhatt et al., 2012*). Libraries were sequenced on an Illumina HiSeq Genome analyzer in the sequencing core at the Broad Stem Cell Research Center at UCLA. Data were analyzed using the standard TopHat, Cufflinks, Cuffmerge, and Cuffdiff pipeline generating 312M and 327M mapped reads from NesKO and wild-type mice respectively at E18 (*Trapnell et al., 2012*). At P1, EmxKO and wild-type samples generated 354M and 317M mapped reads respectively. Alternative exon inclusion levels were determined by additional mapping to an exon duo and trio database using Splicetrap (*Wu et al., 2011*). Gene Ontology Analysis was performed using DAVID (*Huang da et al., 2009*).

## Dissociated neuronal culture

Mouse primary cortical cultures were prepared from E15-16 embryos and primary hippocampal cultures from E18 embryos as described previously (*Boutz et al., 2007*). Briefly, brain tissues were dissected, incubated in 0.1% trypsin in Neural Basal medium (Invitrogen), rinsed with Neural Basal containing 5% fetal bovine serum (FBS), and dissociated mechanically. Dissociated cells were plated onto poly-L-lysine (10µg/ml) coated dishes with Neural Basal supplemented with B27 and GlutaMax (Invitrogen). Cultures were fed with 1/3 volume of fresh Neural Basal/B27/GlutaMax every 3 days. All cultures were fixed with 4% paraformaldehyde (PFA) in phosphate-buffered saline (PBS) for 20 min at room temperature for immunocytochemistry.

Viability Assays (*Figure 4C*) were carried out using the Life Technologies Live/Dead Cell Viability Kit. Briefly, live cells stained with Calcein and dead cells stained with Ethidium dimer were distinguished and counted on multiple microscopic fields to determine the percent of surviving cells at the different time points in culture. For the data in *Figure 4C*, cells from 4 knockout E16-17 embryos derived from two litters, and four wild-type littermate embryos, were plated at $0.5 \times 10^6$ cells/ml onto coated coverslips in 24-well plates. One coverslip from each embryo was stained for viability at each time point, yielding four coverslips for each genotype at each time point (DIV1, 4, 7, 14, 18). Four regions of each coverslip were counted and summed to yield total cell numbers, with the wild type counts on DIV1 averaged to provide the starting number for determination of 'percent survival' at subsequent time points. At DIV1, the four wild-type wells averaged 417 viable cells (SD = 29) and the four knockout wells averaged 442 viable cells (SD = 30).

Datasets are submitted to GEO with Accession Number GSE51740, downloadable at: http://www.ncbi.nlm.nih.gov/geo/query/acc.cgi?acc=GSE51740.

## Acknowledgements

We are particularly grateful to Kyu-Hyeon Yeom for identifying an error in *Figure 1* and generating the data to correct it, as well as optimizing the genotyping. We thank Manny Ares and Lily Shiue for initiating the microarray studies and for ongoing helpful discussions of genomewide analysis. Helpful advice on neural development and phenotypic analysis throughout this project came from other members of the Black Lab, Larry Zipursky, Kelsey Martin, and William Yang, who also gave us the Emx1-cre mice. Manny Ares, Lily Shiue, and Kelsey Martin gave us helpful comments on the manuscript. We thank Robert Darnell and Donny Licatalosi for discussions and for the exchange of unpublished information. DLB is an Investigator of the Howard Hughes Medical Institute.

# Additional information

## Funding

| Funder | Grant reference number | Author |
|---|---|---|
| Howard Hughes Medical Institute | | Douglas L Black |
| Brain & Behaviour Research Foundation (formerly NARSAD) | | Qin Li |
| National Institutes of Health | K99 MH096807, R01 GM084317, R01 GM49662 | Sika Zheng, Douglas L Black |

The funders had no role in study design, data collection and interpretation, or the decision to submit the work for publication.

## Author contributions

QL, SZ, AH, PS, Conception and design, Acquisition of data, Analysis and interpretation of data, Drafting or revising the article; C-HL, Analysis and interpretation of data, Drafting or revising the article; X-DF, Conception and design, Drafting or revising the article; DLB, Conception and design, Analysis and interpretation of data, Drafting or revising the article

## Ethics

Animal experimentation: Animals were maintained by the University of California, Los Angeles (UCLA) Division of Laboratory Medicine, as accredited by the Association for Assessment and Accreditation of Laboratory Animal Care (Assurance #A3196-01). All experiments were approved by the UCLA Animal Research Committee, Protocol #1998-155-43A.

# Additional files

## Major datasets

The following datasets were generated:

| Author(s) | Year | Dataset title | Dataset ID and/or URL | Database, license, and accessibility information |
|---|---|---|---|---|
| Li Q, Zheng S, Han A, Lin C-H, Stoilov P, Fu X-D, Black DL | 2013 | Alternative splicing in PTBP2 knockout mouse brain | GSE51740; http://www.ncbi.nlm.nih.gov/geo/query/acc.cgi?acc=GSE51740 | Publicly available at GEO (http://www.ncbi.nlm.nih.gov/geo/). |
| Li Q, Zheng S, Han A, Lin C-H, Stoilov P, Fu X-D, Black DL | 2014 | Data From: The splicing regulator PTBP2 controls a program of embryonic splicing required for neuronal maturation | http://dx.doi.org/10.5061/dryad.cv39v | Available at Dryad Digital Repository under a CC0 Public Domain Dedication. Supplementary file 1: Exons validated by RT/PCR. A variety of exons showing differences in splicing between wild-type and *Ptbp2* nestin-KO brain were subjected to further analysis by RT/PCR. These included exons identified by MJAY array, RNAseq, and exons previously reported to be PTBP targets. Gene names, exon coordinates, and percent spliced in values (PSI) are listed. Genome coordinates are for mm9. Supplementary file 2: Splicing changes identified by exon junction microarray. RNA from WT and Ptbp2-NestinKO E18 pups was analyzed on Affymetrix MJAY arrays (see 'Materials and methods'). Splicing events yielding Sepscores greater than 0.345 are listed. Supplementary file 3: Splicing changes identified by RNAseq analysis in Ptbp2-NesKO mice. RNAseq reads from E18 WT and NesKO mice were subjected to Splicetrap |

analysis ('Materials and methods'). Events with a change in PSI of greater than 10% are shown with genome coordinates, read numbers, and coverage statistics. Supplementary file 4: Gene Ontology analysis of functional or process enrichment for genes showing splicing changes in Ptbp2-NesKO mice. Enrichments are relative to all genes expressed in wild-type brain. Supplementary file 5: Expression changes in Ptbp2-NesKO mice. Genes showing overall expression changes were identified by Cuffdiff analysis. Supplementary file 6: Gene Ontology analysis of genes up-regulated in Ptbp2-NesKO mice. Genes showing a twofold or greater increase in expression in the KO relative to WT were analyzed for functional or process enrichment relative to all genes expressed in wild-type brain. Supplementary file 7: Gene Ontology analysis of genes down-regulated in Ptbp2-NesKO mice. Genes showing twofold or greater reduced expression in the KO relative to WT were analyzed for functional or process enrichment relative to all genes expressed in wild-type brain. Supplementary file 8: Splicing changes identified by RNAseq analysis in Ptbp2-EmxKO mice. RNAseq reads from P1 WT and EmxKO mice were subjected to Splicetrap analysis ('Materials and methods'). Events with a change in PSI of greater than 10% are shown with genome coordinates, read numbers, and coverage statistics.

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
