## [Decision Letter]

Thank you for sending your work entitled “The splicing regulator PTBP2 controls a program of embryonic splicing required for neuronal maturation” for consideration at *eLife*. Your article has been favorably evaluated by a Senior editor and 3 reviewers, one of whom, Ben Blencowe, is a member of our Board of Reviewing Editors.

The Reviewing editor and the other reviewers discussed their comments before we reached this decision, and the Reviewing editor has assembled the following comments to help you prepare a revised submission.

In this manuscript, Li and colleagues investigate the role of the neuronally-enriched RNA binding protein and splicing regulator PTBP2 in nervous system development in vivo. Using different conditional knockout mouse lines in combination with phenotypic analyses and transcriptome profiling, the authors investigate the contribution of PTBP2 at different stages of development. The results show that PTBP2 regulates programs of alternative splicing events, and that loss of PTBP2 leads to premature derepression of exons that are normally included later in development. Taken together, the data support a model in which PTBP2 coordinates, in conjunction with PTBP1 and likely other splicing regulators, at least two distinct temporal transitions in the landscape of neuronal splicing regulatory networks: one from neural progenitors to differentiating neurons, and the other from differentiating to mature neurons.

While the effects of loss of PTBP2 on neuronal development has been described recently by the Darnell laboratory (Licatalosi et al. G&D, 2012), the results presented by Li and colleagues significantly extend our understanding of the function of this protein since the use of different conditional knockout backgrounds unmasked phenotypes later in development that could not be observed in null mutants. The results thus emphasize the importance of studying the dynamics of RNA regulatory networks in knockout animals at different stages of development, especially for RNA binding proteins that are dynamically regulated such as PTBP2. Although it is unclear which of the many alternative splicing events regulated by PTBP2 are most critical for the observed mutant phenotypes, this study represents an important step towards the future characterization of these variants. The authors are requested to address (by inclusion of experiments or appropriate modification of the text) the following points in a revised manuscript:

1) Figure 1. In order to demonstrate that the authors are analyzing the desired homologously recombined allele and not an insertion (i.e., one arm homologous recombination), they should include a Southern of the -/- (i.e., FLOX/FLOX) mouse DNA.

2) The Western shown in Figure 1 only displays a slice of the gel where full length protein is detected, whereas the full gel should be shown to confirm whether or not a fragment of PTBP2 is expressed that might retain partial or aberrant activity. This is important to know since it could have bearing on interpretations that follow, including the question as to why there are phenotypic differences compared to those observed in the Darnell study.

3) Figure 4. The authors show that the levels of some proteins with roles in synaptogenesis are reduced, yet there are no experiments addressing whether synapses are actually formed (in the right location, the right numbers or at the right time). This could be addressed by performing immunofluorescence microscopy on appropriate sections. As it stands, the title of Figure 4 is not currently supported by the data presented.

4) The evidence presented for “degeneration” is decreased brain size and that 10-20 day old neuronal cultures from mutants do not survive as well as those from wild type animals. These data may indicate stalled or delayed development or “neurodegeneration”. Typically markers for apoptosis and/or microglial infiltration are used to determine whether neurodegeneration is occurring.

5) The authors state that their PTBP2-KO lines display defects in the formation of axonal tracts (Figure 2), postnatal cortical expansion (Figure 3) and synaptogenesis, but only show one or two sections to support the first two claims. The phenotypic analysis should comprise biological replicates (multiple brains for each genotype) and the defects should be quantified (i.e., measurements of cortical thickness, axonal tract areas should be taken).

6) Compared to results reported in the Darnell study, Li et al. do not observe major effects on neurogenesis in their “full null” strain and they attribute this to possible strain background differences or differences in sensitivity of immunofluorescence based phenotypic assays. While these are possible contributors, there are two intriguing (and ultimately more easily testable) possible explanations not mentioned by the authors: (1) The Nestin-Cre based knockout of PTBP2 occurs after the effects of requirement for PTBP2 in neurogenesis. Nestin expression commences upon neuronal differentiation not during the earliest of neurogenic stages; (2) PTBP2 expression in non-neuronal cell types (such as glia, for instance) might contribute to proper neurogenesis. Accordingly, the authors are requested to provide a deeper discussion of possible factors underlying differences between results from the two studies.

[Editors' note: further clarifications were requested prior to acceptance, as described below.]

Thank you for resubmitting your manuscript entitled “The splicing regulator PTBP2 controls a program of embryonic splicing required for neuronal maturation” for further consideration at *eLife*. Your revised article has been favorably evaluated by a Senior editor and a member of the Board of Reviewing Editors. The manuscript has been improved but there are some remaining issues that need to be addressed before acceptance, as outlined below:

Related to your responses to specific comments from the previous review, please address the following.

*Main point*
*3*

The authors were not asked by the reviewers to analyze individual synapses, whereas overall synapse distribution can be examined using immunofluorescence. Visualizing the distribution of synaptic markers in sections reveals whether or not developing neurons lay down rudimentary synapses, and if they do, where this happens. If neurons in culture do not develop to the point where synapses can be examined, then no statement/claims can be made concerning effects on synapse formation/synaptogenesis. A reduction of “synaptic” protein levels could just reflect stalled development/overall differentiation. While the authors have acknowledged and corrected some of the previous “overstatements” related to this point, they are requested to carefully check their manuscript to ensure that the description of the experiments and results are not misleading.

*Main point*
*5*

The authors are requested to at least provide the numbers of mice analyzed in their study (what does “all”, “multiple mice”, “multiple litters” etc mean?). The previous comment was not a request for the authors to breed many additional mice, but rather to provide additional support from their existing data that phenotypes are robust between individual mice. Please also state the number of cells counted in Figure 4.

---

## [Author Response]

*1)*
Figure 1*. In order to demonstrate that the authors are analyzing the desired homologously recombined allele and not an insertion (i.e., one arm homologous recombination), they should include a Southern of the -/- (i.e., FLOX/FLOX) mouse DNA*.

We show a Southern in Figure 1 with a combination of 5' and 3' probes that demonstrates the generation of the expected two-arm recombinant allele in the presence of the wildtype allele. An insertion or one-arm recombinant would not be expected to produce both of the observed bands (10.5 kb and 2.6 kb). Thus, we don’t see what information would be gained with an additional Southern lane. Note that we have also added the PCR primers used for genotyping to the diagram to make the assay in Figure 1 clearer.

*2) The Western shown in*
Figure 1
*only displays a slice of the gel where full length protein is detected, whereas the full gel should be shown to confirm whether or not a fragment of PTBP2 is expressed that might retain partial or aberrant activity. This is important to know since it could have bearing on interpretations that follow, including the question as to why there are phenotypic differences compared to those observed in the Darnell study*.

This point is well taken, considering that the RNAseq data shows continued expression from the PTBP2 locus of RNA lacking the targeted exon 4. In extensive western blots we do not observe any truncated PTBP2 protein products in any of our knockout mice. We now present a Western showing the entire lane as Figure 1—figure supplement 1.

*3)*
Figure 4*. The authors show that the levels of some proteins with roles in synaptogenesis are reduced, yet there are no experiments addressing whether synapses are actually formed (in the right location, the right numbers or at the right time). This could be addressed by performing immunofluorescence microscopy on appropriate sections. As it stands, the title of*
Figure 4
*is not currently supported by the data presented*.

Immunofluorescence microscopy to our knowledge cannot resolve individual synapses in sections of whole brain. To observe individual synapses containing particular markers, these immuno-stains are used on primary cultured neurons. However, the PTBP2^-/-^ cultured neurons do not survive long enough to form synapses, which usually form after about two weeks in culture. Thus staining for them would not be informative. However, the reviewers are correct that the title of the figure was an overstatement. Our discussion of the data for this figure made a misstatement that combined the results from the whole brain and the cell culture. We have rewritten our description of this and corrected the title of the figure to be more judicious in our interpretation.

*4) The evidence presented for “degeneration” is decreased brain size and that 10-20 day old neuronal cultures from mutants do not survive as well as those from wild type animals. These data may indicate stalled or delayed development or “neurodegeneration”. Typically markers for apoptosis and/or microglial infiltration are used to determine whether neurodegeneration is occurring*.

We should have said more about this. The cortices in the EmxKO are not just smaller or stalled in development, but are actively degenerating. We thought this was clear from the presented sections. We have stained sections for various markers and there is clearly widespread cell death. The problem is that the tissue is extremely unstable and difficult to stain, yielding very messy sections, and we see staining for a variety of markers making the results so far rather equivocal for understanding the pathway of cell death. We have added further explanation of this in the Results section.

*5) The authors state that their PTBP2-KO lines display defects in the formation of axonal tracts (*Figure 2*), postnatal cortical expansion (*Figure 3*) and synaptogenesis, but only show one or two sections to support the first two claims. The phenotypic analysis should comprise biological replicates (multiple brains for each genotype) and the defects should be quantified (i.e., measurements of cortical thickness, axonal tract areas should be taken)*.

The phenotypes we discuss were found in all mice of a particular genotype. We have added clarification on the consistency of our findings across multiple mice and have added scale bars to the figures to allow easier comparison across the presented sections. The reviewers are correct that to really add statistical rigor to the phenotypic analysis, one would need to quantify the defects over a large number of mice. This would take at least 18 months of additional mouse breeding. We feel this is beyond the scope of this study, where our intent is to make an initial report of these complex phenotypes – each portion of which will require substantial additional analysis. It is perhaps worth noting that the phenotypes we observe appear to be more penetrant than the ectopic mitoses reported in the Darnell paper described below.

*6) Compared to results reported in the Darnell study, Li et al. do not observe major effects on neurogenesis in their “full null” strain and they attribute this to possible strain background differences or differences in sensitivity of immunofluorescence based phenotypic assays. While these are possible contributors, there are two intriguing (and ultimately more easily testable) possible explanations not mentioned by the authors: (1) The Nestin-Cre based knockout of PTBP2 occurs after the effects of requirement for PTBP2 in neurogenesis. Nestin expression commences upon neuronal differentiation not during the earliest of neurogenic stages; (2) PTBP2 expression in non-neuronal cell types (such as glia, for instance) might contribute to proper neurogenesis. Accordingly, the authors are requested to provide a deeper discussion of possible factors underlying differences between results from the two studies*.

This also needed more explanation. We were not clear that we also did not observe the neurogenesis phenotype reported by Darnell even in our full null mouse. So our different observations are not simply due to the differences between the nestinKO and the full null. While the reviewers are correct that there are markers for neuronal progenitors that are somewhat earlier than Nestin, Nestin expression is induced quite early and is turned off when progenitors exit mitosis to differentiate. These same progenitors give rise to glial cells but this occurs primarily after the developmental period we analyze here. Thus, we don’t think that either of the two explanations offered by the reviewers are likely. Nestin positive progenitors express very low amounts of PTBP2, but the primary period of PTBP2 expression is after the downregulation of nestin with the onset of differentiation. To clarify these questions, we have added Figure 1 to the paper. This shows nestin expression in the progenitor cells of the subventricular zone and PTBP2 expression in the maturing neurons that have migrated to outer layers. This expression pattern is consistent with our observed defect in neuronal maturation. From our reading of the Darnell paper, the phenotype of ectopic mitoses in the subventricular zone appears to be sporadic and not able to explain the large differences in splicing or the neonatal death seen in their and our mice. Nevertheless, the fact that we haven’t yet observed this defect does not mean that we doubt the result. Most splicing factor knockouts analyzed so far have highly pleiotropic phenotypes and the early defect observed and its relation to Numb splicing will be very interesting to examine further.

[Editors' note: further clarifications were requested prior to acceptance, as described below.]

Main point 3

*The authors were not asked by the reviewers to analyze individual synapses, whereas overall synapse distribution can be examined using immunofluorescence. Visualizing the distribution of synaptic markers in sections reveals whether or not developing neurons lay down rudimentary synapses, and if they do, where this happens. If neurons in culture do not develop to the point where synapses can be examined, then no statement/claims can be made concerning effects on synapse formation/synaptogenesis. A reduction of “synaptic” protein levels could just reflect stalled development/overall differentiation. While the authors have acknowledged and corrected some of the previous “overstatements” related to this point, they are requested to carefully check their manuscript to ensure that the description of the experiments and results are not misleading*.

I confess that I didn’t entirely understand this point at first, but upon re-reading the text I agree that we made some statements regarding the data that were not quite correct, or might be interpreted differently than we intended. Very simply, if the cells do not survive long enough to reach the major stage of making synapses then we cannot draw conclusions about the mutation specifically affecting synaptogenesis. Moreover, since some synaptic proteins are being expressed, often with altered splicing, we cannot exclude that early synapses are not forming. I have gone through the text making changes to make this clearer, with particular changes to the Results section, as well as changes to the titles and legends of the figures.

Main point 5

*The authors are requested to at least provide the numbers of mice analyzed in their study (what does “all”, “multiple mice”, “multiple litters” etc mean?). The previous comment was not a request for the authors to breed many additional mice, but rather to provide additional support from their existing data that phenotypes are robust between individual mice. Please also state the number of cells counted in*
Figure 4.

We now report the overall numbers in the Discussion that at least 12 litters were analyzed for each of the three mutant genotypes. For individual analyses, such as the histology and cell survival assays, we report in the legends for Figures 2, 3 and 4 that analyses were done on four mutant mice and compared to four wild type littermate controls from two litters. Regarding the cell counts in Figure 4, we were embarrassed to find that we had not previously described how the viability assays were performed. We now present this in detail in the Materials and methods, and describe briefly in the legend to Figure 4 the requested cell numbers (starting at an average of 417 at DIV1).